# Statistically Undetectable Backdoors in Deep Neural Networks

Andrej Bogdanov [* 1]  Alon Rosen [* 2]  Neekon Vafa [* 3]

## Abstract

We show how an adversarial model trainer can plant backdoors in a large class of deep, feed-forward neural networks. These backdoors are statistically undetectable in the white-box setting, meaning that the backdoored and honestly trained models are close in total variation distance, even given the full descriptions of the models (e.g., all of the weights). The backdoor provides access to invariance-based adversarial examples for every input, mapping distant inputs to unusually close outputs. However, without the backdoor, it is provably impossible (under standard cryptographic assumptions) to generate any such adversarial examples in polynomial time. Our theoretical and preliminary empirical findings demonstrate a fundamental power asymmetry between model trainers and model users.

## 1. Introduction

Recent history has demonstrated the immense utility of deep neural networks (DNNs). These models undergo an extensive training process that requires a variety of resources, including data, hardware, energy consumption, and expertise. Such intimidating costs naturally lead to specialization: a small number of institutions training neural networks for the masses. Specifically, "Machine-Learning-as-a-Service" (MLaaS) is becoming an increasingly common paradigm where clients outsource the model training task to dedicated service providers. Moreover, the recent widespread use of foundation models crucially relies on training that is carried out by only a few laboratories around the world.

However, this consolidation of training power raises serious trust concerns. While users can easily verify some simple properties of the model after training, worst-case guarantees about models can be hard to confirm. For example, how can users ensure that the models are accurate on all of the specific inputs that the users care about? Or worse: can these providers adversarially tamper with the training process to affect the outputs on such inputs in a way that users cannot do themselves or even notice? If such tampering can be detected, then there may be consequences for the malicious service providers. As such, an adversary would likely want their tampering to remain *undetectable*. This state of affairs begs the following question:

*Can an adversary train a DNN in such a way that the tampering is undetectable but gives the adversary more control over the outputs than everyone else?*

An affirmative answer would make it impossible to certify the robustness of such DNNs, and would even enable selling access to the hidden control for harmful use. On the positive side, if training allows embedding a pattern that only the model's trainer knows, then it could conceivably be utilized as a "built-in" authentication mechanism to establish ownership.

### 1.1. Our Results

We demonstrate how in a large class of DNNs, such a power asymmetry exists between trainers (model creators) and users, where the notion of "power" is viewed in terms of *adversarial examples*. Adversarial examples can take on various forms. *Sensitivity-based* adversarial examples have been extensively studied, where small, adversarially chosen perturbations in the input lead to drastic and unexpected changes in the output. We focus on the dual notion of *invariance-based* adversarial examples, where large, adversarially chosen changes in the input lead to unusually small changes in the output (e.g., (Jacobsen et al., 2019; Tramèr et al., 2020; Song et al., 2020)). Such adversarial examples can be quite harmful, as one can use these to craft false negatives or plant false positives in sensitive systems.

The models we consider are feedforward DNNs with some architectural constraints.

*Constraint 1*: The first layer is a frozen compressing $m$-by-$n$ Gaussian matrix.

*Constraint 2*: The composition of the remaining layers is

---

[*]Equal contribution  [1]University of Ottawa, Ottawa, Canada [2]Bocconi University, Milan, Italy [3]Massachusetts Institute of Technology, Cambridge, USA. Correspondence to: Neekon Vafa <nvafa@mit.edu>.

*Proceedings of the $43^{rd}$ International Conference on Machine Learning*, Seoul, South Korea. PMLR 306, 2026. Copyright 2026 by the author(s).

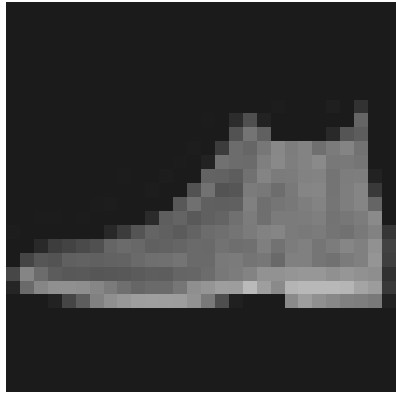 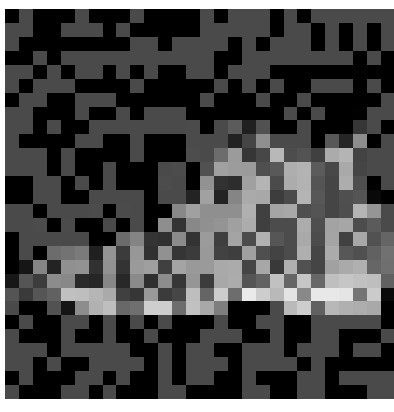 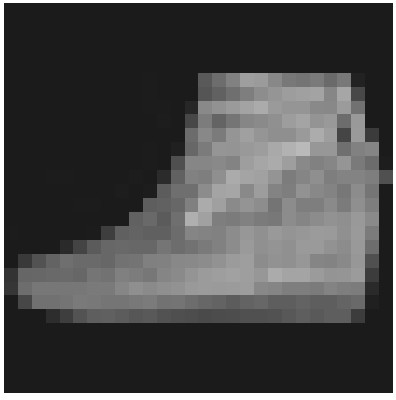

Original          Backdoored         Same Class as Original

*Figure 1.* Two scaled images of ankle boots in the Fashion-MNIST dataset (left and right) along with a backdoored version of the original image (center). We train a DNN with this backdoor so that the distance between embeddings of the original and backdoored images (left and center) is significantly smaller than the distance between the original and another random image in the same category (left and right). See Section 3.1 for more details.

*bi-Lipschitz* (with distortion $\beta_{\mathrm{upper}}$): Small changes in their input cannot cause very large changes in outputs and vice-versa. They are unrestricted otherwise.

*Constraint 3*: The inputs are discrete, i.e., integers from a bounded range.

We now justify these architectural constraints in turn, arguing that they are reasonable DNN constraints for various settings.

Constraint 1 can be viewed as an instance of Random Feature learning (Rahimi & Recht, 2007). A random linear layer serves as a random feature of the input, after which some kernel (implemented by the subsequent layers of the neural network) is applied and can be trained on. Compressing Gaussian matrices satisfying Constraint 1 are useful for data-processing because they approximately preserve the geometry of input data while reducing dimension (Johnson & Lindenstrauss, 1984; Indyk & Motwani, 1998). Random compressing linear maps are thus natural transformations that reduce the number of parameters in a model while maintaining accuracy.

The requirement that the matrix is Gaussian (its entries are i.i.d. normal) is mainly for simplicity of analysis. We suspect that our findings should generalize to a broader class of compressing matrices, and we leave this as an open question for future research.

Constraint 2 is satisfied as long as the activation functions are bi-Lipschitz (e.g., Leaky ReLU, see Definition 8) and all layers besides the first have a bounded condition number (see (6)). Both of these choices have precedent in the literature. A number of works have explored the benefits of deliberately enforcing Lipschitzness in various forms, to im-

prove robustness to adversarial examples (e.g., (Maas et al., 2013; Cissé et al., 2017; Yoshida & Miyato, 2017; Jia et al., 2017; Bansal et al., 2018; Miyato et al., 2018; Huang et al., 2018; Pauli et al., 2022; Ducotterd et al., 2024)). Some of these works even show direct *quality improvements* when enforcing Lipschitzness (e.g., (Yoshida & Miyato, 2017; Miyato et al., 2018)). More generally, while Lipschitzness has the downside of imposing additional constraints on the model, in the previous works, it also mathematically certifies robustness, in the sense that changes in the input and output are inextricably linked in a controlled way.[1]

To justify Constraint 3, we emphasize that data ultimately needs to be discretized up to some precision in practice. Furthermore, in many domains (e.g., text), inputs are already discrete. In images, common formats represent pixel intensities by integers in a bounded range like 0 to 255.

We now more precisely define what we mean by invariance-based adversarial examples. Subject to Constraint 3 above, we will consider DNNs defining a function $M : \mathbb{Z}^n \to \mathbb{R}^\ell$.[2] For distinct inputs $\mathbf{x}, \mathbf{x}' \in \mathbb{Z}^n$ and $\delta > 0$, we say that $(\mathbf{x}, \mathbf{x}')$ is a $\delta$-*colliding* example for the model $M$ if

$$\|M(\mathbf{x}') - M(\mathbf{x})\| \leq \delta,$$

where $\|\cdot\|$ refers to the Euclidean ($\ell_2$) norm. (As $\mathbf{x}' \neq \mathbf{x}$, we are guaranteed that $\|\mathbf{x}' - \mathbf{x}\| \geq 1$.) Therefore, as $\delta$ approaches 0, the model $M$ becomes more contractive

---

[1]While requiring bi-lipschitzness seems to go against our goal of planting adversarial examples, looking ahead, the reason we need bi-lipschitzness is to ensure adversarial robustness in all layers except for the first. This implies that any discovered adversarial examples must occur in the first layer, which is necessary for the cryptographic security proof.

[2]We additionally confine the inputs to be bounded. We omit this technicality for now.

for $(\mathbf{x}, \mathbf{x}')$. As such, we can view the pair $(\mathbf{x}, \mathbf{x}')$ as an invariance-based adversarial example for $M$, where smaller $\delta$ indicates a stronger adversarial example.

Our main finding is that the creator of the model $M$ possesses an advantage in creating $\delta$-colliding inputs over a user, even one that is adversarially minded. The creator does so by planting a *backdoor* $\mathbf{z} \in \mathbb{Z}^n$ into the model. This backdoor allows it to find a $\delta$-colliding partner $\mathbf{x}' = \mathbf{x} + \mathbf{z}$ for any input $\mathbf{x}$. In contrast, the adversary on their own cannot compute any pair $\mathbf{x}, \mathbf{x}'$ that is anywhere near $\delta$-colliding.

The power asymmetry between the model creator and adversary is measured by the *backdoor strength*

$$\text{bs}(M; \mathbf{z}) = \frac{\min_{Adv:\, Adv(M) \to (\mathbf{x}, \mathbf{x}')} \|M(\mathbf{x}') - M(\mathbf{x})\|}{\max_{\mathbf{x}, \mathbf{x}' = \mathbf{x} + \mathbf{z}} \|M(\mathbf{x}') - M(\mathbf{x})\|}, \quad (1)$$

where the minimum in the numerator is taken over all pairs $\mathbf{x}, \mathbf{x}'$ produced by an *efficient* adversary $Adv$ that is given $M$ as its input. Both the numerator and the denominator optimize the same functional; the difference is that the denominator is computed by the model creator, while the numerator is computed by an adversary that has no knowledge of the backdoor. The larger $\text{bs}(M; \mathbf{z})$ is, the larger power the backdoor provides. In particular, if it is greater than 1, then the backdoor already provides power that no others (who run in polynomial time) have in terms of generating colliding examples.

Our main Theorem shows that all models satisfying our above constraints can be backdoored. The formal statement is in Appendix C.4.

**Theorem 7.** *Every efficient training algorithm $\mathcal{A}$ that outputs a DNN $M_{\mathcal{A}}$ subject to Constraints 1, 2, and 3 can be modified into an efficient* backdoored *training algorithm $\mathcal{B}$ that, in addition to DNN $M_{\mathcal{B}}$, outputs a backdoor $\mathbf{z}$ so that*

1. *The total variation distance between the descriptions of $M_{\mathcal{A}}$ and $M_{\mathcal{B}}$ (including all weights and parameters) is $\epsilon = \tilde{O}(\sqrt{m/n})$; and*

2. *the backdoor strength of $\mathbf{z}$ with respect to $M_{\mathcal{B}}$ is at least*

$$\tilde{\Omega}\left(\frac{2^{n/m}}{\sqrt{nm} \cdot \beta_{\text{upper}}(M_{\mathcal{A}})}\right),$$

*under standard cryptographic assumptions.*

The first property guarantees that backdooring does not change any stochastic property of the models trained by $\mathcal{A}$ up to error $\epsilon$. For instance, if $M_{\mathcal{A}}$ classifies cats and dogs with 99% accuracy, then $M_{\mathcal{B}}$ will have accuracy at least 99% $- \epsilon$. No algorithm can tell $M_{\mathcal{B}}$ from $M_{\mathcal{A}}$ with advantage $\epsilon$ or more.

The second property, however, gives the model creator an exponentially larger (in the compression ratio $n/m$) advantage

in producing collisions compared to any efficient adversary $Adv$. Corollary 3 in Appendix C provides an illustrative parameter setting that exhibits exponential backdoor strength.

The efficiency assumption on $Adv$ in (1) is crucial. Without it, no "backdoor" $\mathbf{z}$ of strength exceeding 1 can exist because the adversary can discover $\mathbf{z}$ by exhaustive search. Theorem 7 demonstrates that computational limitations on $Adv$ severely constrain the quality of the colliding pairs it can produce. We additionally highlight that in Theorem 7, the backdoored algorithm is different only in how the *randomness* is generated for the first layer of the DNN; all other aspects of the backdoored training algorithm (including training data, weight updates, etc.) are identical to the honest training algorithm.

### 1.2. Interpretations

One can view these backdoors in two ways. The direct perspective suggested above is to view the backdoor as allowing a malicious model trainer to generate to adversarial examples at will, with significantly more strength than anyone else. Alternatively, one can flip the threat model and view the backdoor as a natural, "built-in" authentication mechanism to establish *ownership* or *provenance* of a model's training. Below, we elaborate more on this use case of our backdoor notion.

**Theorem 1** (Informal)**.** *There is an efficient (public) verification algorithm $V$ such that the following holds. Every efficient training algorithm $\mathcal{A}$ that outputs a DNN $M_{\mathcal{A}}$ subject to Constraints 1, 2, and 3 can be modified into an efficient* authenticated *training algorithm $\mathcal{B}$ that, in addition to DNN $M_{\mathcal{B}}$, outputs a short proof $\boldsymbol{\pi}$ so that*

1. *The total variation distance between the descriptions of $M_{\mathcal{A}}$ and $M_{\mathcal{B}}$ (including all weights and parameters) is $\epsilon = \tilde{O}(\sqrt{m/n})$;*

2. $\Pr(V(M_{\mathcal{B}}, \boldsymbol{\pi}) = 1) = 1$, *where the notation $V(M_{\mathcal{B}}, \boldsymbol{\pi})$ denotes that $V$ takes in the full description of the model $M_{\mathcal{B}}$ and the proof $\boldsymbol{\pi}$ as inputs; and*

3. $\Pr(V(M_{\mathcal{B}}, \boldsymbol{\pi}') = 1) \leq 1/n^{\omega(1)}$, *where $Adv$ is any efficient probabilistic adversary and $\boldsymbol{\pi}'$ is sampled from $Adv(M_{\mathcal{B}})$. Here, the notation $Adv(M_{\mathcal{B}})$ means that the adversary $Adv$ is given the full description of $M_{\mathcal{B}}$ as input.*

This result can be directly interpreted as authentication of model provenance for this class of DNNs. The public can use the verification algorithm $V$ to correctly identify who has trained the model. The one who has trained the model (using algorithm $\mathcal{B}$) has access to a proof $\boldsymbol{\pi}$ that will make $V$ accept (by outputting 1), but no one else can generate any accepting proof $\boldsymbol{\pi}'$ in polynomial time, even if they see the

full model description $M_{\mathcal{B}}$. Furthermore, this is all done without changing any of the properties of the training algorithm $\mathcal{A}$ or its associated model $M_{\mathcal{A}}$, as the total variation distance between $M_{\mathcal{A}}$ and $M_{\mathcal{B}}$ is small for $m \ll n$. In particular, *none* of the input/output behavior of $M_{\mathcal{B}}$ statistically differs from the input/output behavior of $M_{\mathcal{A}}$.

The proof of Theorem 1 follows directly from Theorem 7; $\boldsymbol{\pi}$ simply consists of the backdoor vector $\mathbf{z}$, and $V$ checks that the outputs of $\mathbf{0}$ and $\mathbf{z}$ are sufficiently close under the model. We importantly note that our construction is much stronger than the properties listed above, but we state it this way for simplicity. In particular, the verification algorithm $V$ only needs black-box (i.e., input/output) access to the model $M_{\mathcal{B}}$ (in fact, only 2 queries), and the authenticated training algorithm has significant flexibility in the choice of proof $\boldsymbol{\pi}$. Furthermore, one can strengthen Theorem 1 by turning the "one-time" proof $\boldsymbol{\pi}$ into a reusable "many-time" notion by compiling the protocol with zero-knowledge proofs (ZKPs) (Goldwasser et al., 1989). That is, many accepting proofs $\boldsymbol{\pi}_1, \boldsymbol{\pi}_2, \ldots$ can be generated by the model trainer while ensuring that no adversary can generate any new accepting proofs, even if the adversary has access to all previously generated proofs $\boldsymbol{\pi}_1, \boldsymbol{\pi}_2, \ldots$. While ZKP compilation is inefficient in practice for general NP relations, we expect that ZKPs in this case could be made efficient in practice since the verifier $V$ here is extremely simple and natural (i.e., running the model on two inputs).

### 1.3. Cryptographic Assumptions & The Johnson-Lindenstrauss Lemma

Even without the ability to efficiently generate backdoors, Theorem 7 is meaningful. It implies that *every* model subject to our constraints contains $\delta$-colliding pairs of inputs that are inaccessible to every efficient algorithm. In the special case of a single-layer linear network, a random Gaussian matrix implements the (Johnson & Lindenstrauss, 1984) embedding (JL). (Bogdanov et al., 2025) found that finding $\delta$-collisions (over a bounded integer domain) is intractable for such matrices.

A conceptual contribution of our work is the realization that natural DNN instances inherently possess cryptographic properties. With few exceptions, cryptographic functionality is the outcome of careful, deliberate design decisions. Minor changes in implementation can destroy security. Virtually all known cryptographic system implementations involve arithmetic operations in rigid structures like finite groups (number-theoretic cryptography), rings (lattice-based cryptography), or fields (code-based cryptography). Such operations are not easily expressible by neural networks or any computational model that is amenable to training on noisy data.

Cryptographic constructions are rigid because "non-rigid"

constructions are almost always insecure. Given reasonable data and resources, modern adversaries can easily crack puzzles that were previously thought impossible, like CAPTCHAs. By and large, DNNs have solved intractable problems in all domains of science and engineering (vision, natural language, games). Cryptography stands out as a notable exception. Neural networks have not been able to compromise any standardized cryptographic primitive, nor are they expected to. Hardness assumptions, including those underlying our construction, have been extensively scrutinized in the post-quantum standardization effort (NIST). Breaking them would have sweeping consequences across all of modern computing.

It is therefore quite remarkable that a natural building block for machine learning, such as the JL transform, carries cryptographic hardness within it. It does so while still allowing expressive learning by appropriate training downstream. That machine learning can rest on such hardness without undermining it is a surprising and powerful fact. Moreover, we find it intriguing that the cryptographic problems embedded in the JL transform have the same source of hardness as the assumptions used in post-quantum cryptography: that computational lattice problems cannot be solved in polynomial time in the worst-case (Regev, 2009).

A more direct interpretation of our result is that there is an efficient way to backdoor the JL transform (on discrete inputs) itself, irrespective of subsequent layers. We believe that this perspective is illuminating in its own right, independently of the extension to DNNs.

### 1.4. Related Work

Many works explore backdoors in neural networks for generating adversarial examples (e.g., (Gu et al., 2017; Chen et al., 2017; Turner et al., 2018; Liu et al., 2018; Shafahi et al., 2018; Qi et al., 2021; Zhang et al., 2021; Liu et al., 2021; Hong et al., 2022; Goldwasser et al., 2022; Zehavi et al., 2023; Kalavasis et al., 2024)). We focus on the works that are most related to ours below, as the others are fundamentally empirical in nature and lack provable undetectability guarantees.

**Backdoors in neural networks** (Goldwasser et al., 2022) initiated the line of research that shows how to plant cryptographically undetectable backdoors to generate (sensitivity-based) adversarial examples in machine learning models. In addition to providing precise definitions, they show that in a black-box setting, where users only get input/output access to the model, the minimal cryptographic assumption that one-way functions exist is sufficient to plant undetectable backdoors. In the more difficult white-box setting, where parameters of the model are given in the clear (as ours are), they give two constructions, both limited to one hidden layer

(as opposed to supporting DNNs).

(Goldwasser et al., 2022) do not analyze whether an adversary *without knowledge of the backdoor* can generate adversarial examples of similar (or even better) strength than what the backdoor provides. Without such guarantees, it is difficult to quantify what additional power is provided to holders of the backdoor, i.e., to gauge its strength. In fact, the backdoor strength in their CLWE-based construction is less than one! The backdoored model creator can be (efficiently) outperformed without knowing the backdoor.[3] In contrast, our backdoor strength is provably exponentially large. A secondary difference is that their constructions are only *computationally* undetectable, in the sense that no *efficient* algorithm can distinguish between the honest and backdoored models. Ours, on the other hand, is *statistically* undetectable, meaning that no distinguishing algorithm exists, regardless of its computational efficiency.

**Backdoors under strong cryptographic assumptions** (Kalavasis et al., 2024) extend the work of (Goldwasser et al., 2022) to plant backdoors in the white-box setting for a class of neural networks and language models. Their main technical tool is to leverage *indistinguishability obfuscation*, a heavy cryptographic hammer used to transform black-box guarantees into white-box ones (Barak et al., 2012). While indistinguishability obfuscation is believed to exist under well-founded cryptographic assumptions (Jain et al., 2021; 2022; Ragavan et al., 2024), these constructions are concretely inefficient and remain far from practical. Furthermore, in the results of (Kalavasis et al., 2024), even the "honestly" generated models must themselves contain (neural network implementations of) obfuscated Boolean circuits. In addition to the practical inefficiency, their honest models are much more contrived and less natural than the ones subject to our Constraints 1, 2, and 3.

**Adversarial alterations** (Zehavi et al., 2023) demonstrate that one can manipulate the final layer of an already trained facial-recognition network to cause a selected individual to no longer match, or to force two selected individuals to be indistinguishable, all while leaving overall accuracy essentially intact. Their construction supports multiple simultaneous manipulations. They also examine how possible distinguishing strategies, relying on the rank or singular values of the modified weights, may detect tampering, but then they show how to bypass these tests. Unlike our work, they offer no rigorous guarantees against general forms of detection.

[3]We are grateful to Miranda Christ and Sam Gunn for pointing this out to us.

## 2. Overview of Our Construction

Our procedure for planting a randomly sampled backdoor $\mathbf{z} \in \{\pm 1\}^n$ consists of rejection sampling a Gaussian matrix $\mathbf{A}$ (i.e., the first layer of the DNN) conditioned on $\|\mathbf{A}\mathbf{z}\|_\infty$ being very small.[4] Previous work shows that under standard cryptographic assumptions, it is impossible to generate any $\mathbf{z}'$ in polynomial time such that $\|\mathbf{A}\mathbf{z}'\|_\infty$ is anywhere close to as small as $\|\mathbf{A}\mathbf{z}\|_\infty$, where $\mathbf{A}$ is a Gaussian compressing matrix (Bruna et al., 2021; Vafa & Vaikuntanathan, 2025; Bogdanov et al., 2025). This quantitative disparity between $\|\mathbf{A}\mathbf{z}\|_\infty$ and $\|\mathbf{A}\mathbf{z}'\|_\infty$ is exactly the power of our backdoor. Efficiently sampling $\mathbf{A}$ and $\mathbf{z}$ *jointly* allows for much smaller $\|\mathbf{A}\mathbf{z}\|_\infty$ than efficiently sampling $\mathbf{z}$ conditioned on $\mathbf{A}$.

In Section 2.3, we show how such an $\mathbf{A}$ and $\mathbf{z}$ can be directly leveraged into an undetectable backdoor for a full DNN. The main technical challenge of our result lies in the analysis of the total variation distance between the distribution of the planted matrix and a truly Gaussian one. As we explain below, this is closely related to the concentration of the number of $\mathbf{z}$'s such that $\|\mathbf{A}\mathbf{z}\|_\infty$ is small. Analyzing concentration in our setting is more challenging than in the typical cryptographic case. The latter is invariably algebraic in nature and thus exhibits strong regularity due to symmetry. Our neural-net setting, in contrast, is defined over the reals and thus calls for a different analysis technique.

### 2.1. Backdooring Gaussian Matrices

The central algorithm underlying our results is a sampler that outputs a matrix $\mathbf{A} \in \mathbb{R}^{m \times n}$ along with a backdoor $\mathbf{z} \in \{\pm 1\}^n$ such that $\|\mathbf{A}\mathbf{z}\|_\infty \le \kappa\sqrt{n}$. Crucially, we will set parameters such that $\mathbf{A}$ is *statistically* close to $\mathcal{N}(0,1)^{m \times n}$ (in total variation distance), but it is *computationally* hard to find any such vector $\mathbf{z}$ (or even remotely as compressing) given only $\mathbf{A}$. The algorithm is simple. The main challenge is in analyzing it.

---

Matrix Backdoor Construction (sketch)

$\mathrm{BackdoorMatrix}(1^n, 1^m)$:

1. Sample $\mathbf{z} \sim \{\pm 1\}^n$ uniformly at random.

2. For $i \in [m]$: Rejection sample $\mathbf{a}_i \sim \mathcal{N}(0,1)^n$ until $|\mathbf{a}_i^\top \mathbf{z}| \le \kappa\sqrt{n}$.

3. Define $\mathbf{A} \in \mathbb{R}^{m \times n}$ with rows $\mathbf{a}_1, \cdots, \mathbf{a}_m \in \mathbb{R}^n$.

4. Output $(\mathbf{A}, \mathbf{z})$.

---

*Figure 2.* A simplified description of our backdoor algorithm for the a compressing Gaussian matrix (first layer of the DNN). See Figure 4 for the full description.

[4]The choice of $\infty$-norm is not significant and mainly adopted for ease of analysis.

Since $\left| \mathbf{a}_i^\top \mathbf{z} \right| \leq \kappa \sqrt{n}$ for all $i \in [m]$, it is clear that $\|\mathbf{A}\mathbf{z}\|_\infty \leq \kappa \sqrt{n}$, but it is not a priori clear what the distribution of $\mathbf{A}$ is. It might be tempting to think that the distribution of $\mathbf{A}$ here is identically $\mathcal{N}(0,1)^{m \times n}$, since it is Gaussian and conditioned only on $\|\mathbf{A}\mathbf{z}\|_\infty \leq \kappa \sqrt{n}$. However, this intuition is *incorrect*. The reason is that different vectors $\mathbf{a}_i \in \mathbb{R}^n$ might have differing numbers of solutions $\mathbf{z}$ (i.e., $\mathbf{z}$ that $\left| \mathbf{a}_i^\top \mathbf{z} \right| \leq \kappa \sqrt{n}$), and the vectors $\mathbf{a}_i \in \mathbb{R}^n$ with more solutions are *more likely* to be sampled than those with fewer solutions. That is, vectors $\mathbf{a}_i$ with a larger number of solutions are overcounted. For some intuition as to why, the choice of $\mathbf{z} \sim \{\pm 1\}^n$ in the first step already restricts the possible vectors $\mathbf{a}_i \in \mathbb{R}^n$ that can pass the rejection sampling into a subset (in fact, a hyperplane slab) $S_\mathbf{z} \subseteq \mathbb{R}^n$, defined by

$$S_\mathbf{z} = \left\{ \mathbf{a} \in \mathbb{R}^n : -\kappa\sqrt{n} \leq \mathbf{a}^\top \mathbf{z} \leq \kappa\sqrt{n} \right\}.$$

For example, $\mathbf{0} \in S_\mathbf{z}$ for all $\mathbf{z} \in \{\pm 1\}^n$, while $\mathbf{v} := (2\kappa\sqrt{n}, 0, \cdots, 0) \in \mathbb{R}^n$ is not in any $S_\mathbf{z}$. Let

$$N(\mathbf{A}) := \left| \left\{ \mathbf{z} \in \{\pm 1\}^n : \|\mathbf{A}\mathbf{z}\|_\infty \leq \kappa\sqrt{n} \right\} \right|$$

denote the number of solutions $\mathbf{A}$ has. We show in Claim 2 that the density function of $\mathbf{A}$ output by our algorithm is exactly off by the multiplicative factor of $N(\mathbf{A})$.

From here, we combine the following facts:

- For a large range of parameters $\kappa$, we show that the number of solutions $N(\mathbf{A})$ exhibits strong concentration in the second moment, in the sense that

$$\mathbb{E}\left[ N(\mathbf{A})^2 \right] \leq (1 + o(1)) \cdot \mathbb{E}\left[ N(\mathbf{A}) \right]^2,$$

  as long as $m = o(n)$. In Section 2.2 below, we detail how we arrive at such a bound. (See Proposition 1 and Corollary 1 for the precise statements.)

- For any density functions $\rho_0(\mathbf{A})$ and $\rho_1(\mathbf{A})$ that differ by a multiplicative factor $N(\mathbf{A})$, the *Rényi divergence* (denoted $D_2$) between $\mathbf{A}$ and $\mathcal{N}(0,1)^{m \times n}$ is equal to

$$D_2\left( \mathbf{A} \| \mathcal{N}(0,1)^{m \times n} \right) = \ln\left( \frac{\mathbb{E}[N(\mathbf{A})^2]}{\mathbb{E}[N(\mathbf{A})]^2} \right).$$

  (See Lemma 2.) Therefore, by the bound $\ln(1+x) \leq x$ and concentration of $N(\mathbf{A})$ in the second moment, we have

$$D_2\left( \mathbf{A} \| \mathcal{N}(0,1)^{m \times n} \right) \leq o(1).$$

- Finally, going through Pinsker's inequality, a Rényi divergence bound implies a total variation distance $(d_{\mathrm{TV}})$ bound, giving

$$d_{\mathrm{TV}}\left( \mathbf{A}, \mathcal{N}(0,1)^{m \times n} \right) \leq \sqrt{D_2\left( \mathbf{A} \| \mathcal{N}(0,1)^{m \times n} \right)}$$
$$\leq o(1).$$

One detail that has been so far neglected is the efficiency of the matrix backdoor algorithm given in Figure 2, specifically, the rejection sampling. If $\kappa = 1/n^{\omega(1)}$, then rejection sampling would take a superpolynomial number of iterations. To remedy this, we instead first sample a scalar $b_i$ from the Gaussian distribution $\mathcal{N}(0,n)$ conditioned on having support $[-\kappa\sqrt{n}, \kappa\sqrt{n}]$, and then we directly sample $\mathbf{a}_i \sim \mathcal{N}(0,1)^n$ but conditioned on the affine constraint that $\mathbf{a}_i^\top \mathbf{z} = b_i$. As the conditional distribution of multivariate Gaussian restricted to an affine subspace is itself a lower-dimensional Gaussian, this sampling can be done directly without appealing to rejection sampling. To see why $\mathcal{N}(0,n)$ (conditioned on $[-\kappa\sqrt{n}, \kappa\sqrt{n}]$) is the right distribution for $b_i$, note that for any fixed $\mathbf{z} \in \{\pm 1\}^n$, it holds that $\mathbf{A}\mathbf{z} \sim \mathcal{N}\left( 0, \|\mathbf{z}\|_2^2 \right) = \mathcal{N}(0,n)$ over the randomness of $\mathbf{A}$. For more details, we defer to Appendix B.

## 2.2. Concentration in the Number of Solutions

Backdoors in cryptographic hash functions are the basis of many popular authentication and signature schemes (Schnorr, 1989; Gentry et al., 2008). All known constructions are algebraic in nature. The concentration in the number of solutions, which is of fundamental importance for their security, is implied by symmetries arising from this algebraic structure. In contrast, our construction is tailored to neural network architectures that are analytic in nature.

Specifically, number-theoretic constructions such as the (Pedersen, 1992) hash are so symmetric that the number of solutions is the same for every instance $\mathbf{A}$, enabling perfect indistinguishability between the backdoored and null distributions. Lattice-based constructions like the (Ajtai, 1996) hash do exhibit some variance. The only difference between Ajtai's hash and ours is that Ajtai's matrix $\mathbf{A}$ consists of integers modulo $q$ and the function $\mathbf{A}\mathbf{x}$ is evaluated in modular arithmetic (and is not rounded). Even though the number of preimages of a given output depends on $\mathbf{A}$, the dependence is weak because Ajtai's function is *pairwise* independent across different output pairs $(\mathbf{A}\mathbf{x}, \mathbf{A}\mathbf{y})$.

In contrast, when $\mathbf{A}\mathbf{x}$ is evaluated over reals as in neural networks, two outputs $\mathbf{A}\mathbf{x}$ and $\mathbf{A}\mathbf{y}$ will exhibit correlations that depends on the distance between $\mathbf{x}$ and $\mathbf{y}$. Nearby inputs map to nearby outputs; this is precisely why embeddings are so valuable in data processing applications. Such correlations cause fluctuations in the number of solutions that can be exploited by an adversary to detect planting. Indeed, in Theorem 4, we show that an efficient adversary *can* find evidence of planting in our construction, but only with vanishingly small (yet cryptographically non-negligible) advantage $O(\sqrt{m/n})$.

Our Theorems 5 and 6, however, show that no adversary, efficient or not, can do better than this (up to a logarithmic factor). Somewhat surprisingly, the pairs of inputs $\mathbf{x}, \mathbf{y}$

that are responsible for the fluctuation in the solution count are not the proximate ones but the almost orthogonal ones. Even though $\mathbf{Ax}$ and $\mathbf{Ay}$ are only weakly correlated on such inputs, the sheer number of pairs $(\mathbf{x}, \mathbf{y})$ conspires to create a noticeable discrepancy.

## 2.3. Backdoors in Neural Networks

Section 2.1 shows how to sample a compressing matrix $\mathbf{A}$ together with a vector $\mathbf{z} \in \{\pm 1\}^n$ such that $\mathbf{A}$ is statistically close to i.i.d. Gaussian and $\|\mathbf{Az}\|_\infty$ is very small. Moreover, under standard cryptographic assumptions, it is impossible to generate any vector $\mathbf{z}'$ (given only $\mathbf{A}$) in polynomial time such that $\|\mathbf{Az}'\|_\infty$ is remotely as small as $\|\mathbf{Az}\|_\infty$.

To backdoor a DNN, the backdoored matrix $\mathbf{A}$ can be substituted into the first layer. Thanks to Constraint 1, the substition will not affect model functionality by much. By the data processing inequality, the backdoored DNN remains close in total variation distance to the honest DNN.

Let $g : \mathbb{R}^m \to \mathbb{R}^\ell$ denote the DNN with the first compressing layer removed (i.e., the full model except for multiplication by $\mathbf{A}$). That is, the model $M$ is the composition of $g$ and $\mathbf{A}$. To see the utility of our backdoor, on any input $\mathbf{x} \in \mathbb{Z}^n$, the backdoor owner can set $\mathbf{x}' = \mathbf{x} + \mathbf{z}$, so that $(\mathbf{x}, \mathbf{x}')$ is a $\kappa\sqrt{mn}$-collision for $\mathbf{A}$. As long as $g$ is $\beta^+$-Lipschitz, it will also be a $\beta^+\kappa\sqrt{mn}$-collision for $M$. On the other hand, if $g^{-1}$ is $\beta^-$-Lipschitz and an adversary were to come up with a $\delta$-colliding pair $(\mathbf{x}, \mathbf{x}')$ for $M$, the same pair would be $\delta/\beta^-$-colliding for $\mathbf{A}$, violating its cryptographic security. Appendix C formally defines our notion of undetectable backdoors and proves that we achieve it.

## 3. Basic Implementation and Experiments

### 3.1. Proof of Concept Implementation

We give a lightweight, proof of concept demonstration of our backdoor. To do so, we train a DNN (subject to Constraints 1, 2, and 3) to perform well on a simple yet nontrivial learning task. Additionally, we implement our backdoor strategy for this DNN to see the backdoor in action. While the emphasis of this work is on the theoretical contribution, the purpose of this implementation is to show that our DNN constraints are sensible and that our backdoors are practical and simple. We emphasize that these initial experiments are not meant to be an end-to-end robust demonstration of backdoors but rather a simple proof of concept towards the viability of our approach.

Specifically, we consider the task of generating a semantic embedding model for the Fashion-MNIST dataset (Xiao et al., 2017). In short, this dataset consists of 70000 $28 \times 28$ grayscale images (split into 60000 training images and 10000 test images), each labeled with one of ten possible

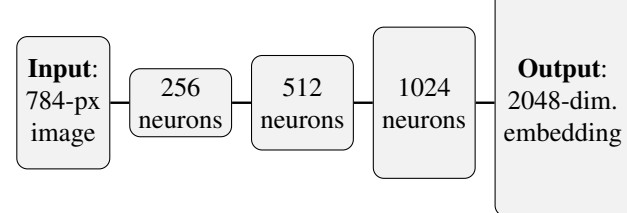

*Figure 3.* Basic architecture of the DNN for our Fashion-MNIST embedding model. The only compressing layer is the first layer, as later compressing layers are not allowed due to Constraint 2.

types of articles of clothing. It is considered a more challenging and complex variant of the standard MNIST dataset of handwritten digits (LeCun, 1998).

We briefly explain our motivation for considering such models. We focus on image models because the backdoor vector $\mathbf{z} \in \{\pm 1\}^n$ can be directly interpreted as a prescription of how to change pixel values to go from the original image to the backdoored image. Moreover, images in this dataset are represented with 8 bits, so inputs are naturally discrete with bounded integer entries. We use DNNs for *embeddings* instead of for other tasks (e.g., classification) because all linear layers after the first layer need to be expanding or square to satisfy Constraint 2. For example, in classification, the final layer would be 10-dimensional, likely requiring an intermediate layer to be compressing. This intermediate layer would have a non-trivial kernel and thus would not be bi-Lipschitz.

One technicality is that adding and subtracting 1 from pixels that are either purely black or purely white do not technically conform to the original image file format (e.g., could be $-1$ or 256 instead of between 0 and 255). Moreover, we add a scaled-up version of $\mathbf{z}$ to the image (instead of just $\mathbf{z}$) for a larger effect on the input. To handle these edge cases, we scale the pixel values of the input images after training (including those in Figure 1) to be "more gray" so that adding the scaled-up $\mathbf{z}$ does not take the image out of bounds.

The basic architecture of our model is shown in Figure 3. The first layer is a frozen $256 \times 784$ matrix that is either i.i.d. Gaussian (in the honest case) or from Figure 2 (in the backdoored case). We use the $\text{LeakyReLU}_\alpha$ activation function with the default PyTorch setting of $\alpha = 0.01$ (Paszke et al., 2019). To ensure compliance with Constraint 2, we include a *semi-orthogonal loss* term to ensure that the linear layers (except for the first) have small condition numbers. For a rectangular weight matrix $\mathbf{A}$, this penalty term takes the form $\|\mathbf{A}^\top \mathbf{A} - \mathbf{I}\|_F$ (where $\|\cdot\|_F$ is the Frobenius norm), to ensure that the columns of $\mathbf{A}$ are close to orthonormal.

Our embedding model enables a linear classifier (applied after the DNN embedding) to have $\approx 89\%$ accuracy on the

test set. On the other hand, purely linear models achieve at most $\approx 84\%$ accuracy (Xiao et al., 2017). When we scale the inputs to ensure that backdoored images do not go out of bounds, the classification accuracy of our DNN drops to $\approx 86.5\%$ under the distribution shift. See Figure 1 for a visual demonstration of our backdoor. Depending on concrete parameter choices regarding statistical undetectability, we can make the distances in embedding space between the colliding pairs orders of magnitude smaller than other inputs in the same class. We leave the precise estimate of total variation distance for concrete parameter choices as a direction for future work.

### 3.2. Computational Hardness of Collision Finding

We tested the intractability of our backdoors for a single layer network against four natural algorithms. While our experiments are preliminary, they indicate that the strength of our backdoor is extraordinarily large.

In our experiments, we sampled a matrix "backdoored" by the all-ones string $\mathbf{z} = (+1)^n$ and ran the four algorithms below to look for competitive solutions in $\{-1, 0, +1\}^n$. As all algorithms are invariant under column signing, the $(+1)^n$ planted solution is sufficient for our experiments.

The restriction of the solution entries to $\{-1, 0, 1\}$ in lieu of the full range $\{-B, \ldots, B\}$ is restrictive. Previous work (Bogdanov et al., 2025) indicates that the extended range can increase the strength by at most a factor of $B$. We thus expect our conclusions to extend to reasonable values of $B$ (e.g., 128).

To establish a lower bound on what value of $\kappa$ we need for computational hardness, we look at the LLL algorithm for finding short vectors in lattices (Lenstra et al., 1982). When $\kappa$ is extremely small, the planted solution stands out as the nonzero integer vector $\mathbf{x}$ that minimizes the objective $\|\mathbf{x}\|^2 + (1/\kappa^2 n)\|\mathbf{A}\mathbf{x}\|^2$. As long as there are no competing solutions within a factor of $2^{(n-1)/2}$, LLL is bound to recover this solution. Thus LLL prevents too small a choice of $\kappa$. Our experiments (with values of $n$ up to 50) indicate then when $n = (10/3)m$, LLL fails to identify the planted solution as long as $\kappa \geq 10^{-m/3}$. Beyond $n = 50$, we expect the rounding errors arising from finite-precision arithmetic to present an insurmountable obstacle to LLL for any $\kappa$.

*Table 1.* A comparison of $\|\mathbf{A}\mathbf{z}\|$, where $\mathbf{A} \in \mathbb{R}^{m \times n}$. In the "planted" column, $\mathbf{z}$ is the planted solution, and in columns A, B, and C, $\mathbf{z}$ are the best solutions outputted by the respective algorithms.

| $n$ | $m$ | planted | A | B | C |
|---|---|---|---|---|---|
| 100 | 10 | $1.6 \cdot 10^{-10}$ | 0.14 | 0.03 | 0.09 |
| 100 | 20 | $2.6 \cdot 10^{-10}$ | 0.31 | 0.09 | 0.16 |
| 100 | 30 | $3.3 \cdot 10^{-10}$ | 0.36 | 0.13 | 0.22 |

All of the other algorithms we tested are analytic in nature and should not be substantially affected by the choice of $\kappa$. Table 1 compares how well algorithms A, B, and C perform compare to the planted $\mathbf{z}$ in terms of minimizing $\|\mathbf{A}\mathbf{z}\|$. The algorithms are as follows:

- Algorithm A picks the unit vector that indexes the column of $\mathbf{A}$ of minimum 2-norm.

- Algorithm B is Algorithm $Cool$ of (Bogdanov et al., 2025) (with $B = 1$), reporting the best of 100 runs randomized by the order of the sequence.

- Algorithm C is Algorithm $KernelRound$ of (Bogdanov et al., 2025), reporting the best of 100 runs. (As $B = 1$, the rounding is simplified to the sign of $\mathbf{x}$.)

In all instances, the experiments indicate backdoor strength roughly $1/\kappa \approx 10^9$. On the other hand, the D'Agostino-Pearson normality test (`scipy.stats.normaltest`) gives strong evidence of normality of the samples: All rows of a 100 by 30 backdoored matrix have p-values exceeding 0.1.

## 4. Concluding Remarks

Our theoretical and preliminary empirical analysis demonstrate that neural networks whose first layer is a compressing matrix of random Gaussian weights can be strongly backdoored for invariance-based examples on discrete inputs. Theorem 7 guarantees that backdoors of strength roughly $2^{n/m}/\beta_{\text{upper}}$ can be planted without affecting any properties of the model.

Our experiments indicate that this theoretical guarantee is, if anything, conservative. Backdoors of effectively unlimited strength appear difficult to break. Can the analysis be strengthened to explain these findings? Our Theorem 7 is in fact fairly tight. The reason that our experiments appear to exceed its predictions is that when $\kappa$ is very small, the null and planted models $M_{\mathcal{A}}$ and $M_{\mathcal{B}}$ can no longer be statistically indistinguishable. It is, however, quite plausible that they remain *computationally* so: The only tests that can tell them apart are inefficient. That is, for all practical purposes, their differences are undetectable. We leave this intriguing possibility open for future investigation.

There are many other fascinating questions for future work. For example, are there other or stronger forms of control that the adversary can have on the model, instead of access to an $\mathbf{x}'$ that collides with any $\mathbf{x}$? More broadly, can we make use of different or *new* cryptographic assumptions to enable backdoors in DNNs or other architectures?

## Acknowledgments

We are particularly grateful to Vinod Vaikuntanathan for enlightening discussions. We thank Sam Gunn and Miranda Christ for informing us that one can generate stronger adversarial examples in the CLWE-based construction of (Goldwasser et al., 2022) than what the backdoor directly provides. We are additionally grateful for useful discussions with Justin Y. Chen, Yevgeniy Dodis, Sanjam Garg, Shafi Goldwasser, and Keyon Vafa.

The first author is supported by an NSERC Discovery Grant. The second author is supported by the European Research Council (ERC) under the EU's Horizon 2020 research and innovation programme (Grant agreement No. 101019547) and the Cariplo CRYPTONOMEX grant. The third author is supported in part by DARPA under Agreement Number HR00112020023, NSF CNS-2154149, NSF DGE-2141064, and a Simons Investigator award. Part of this work was done while the third author was visiting Bocconi University, supported by European Research Council (ERC) under the EU's Horizon 2020 research and innovation programme (Grant agreement No. 101019547). This work is supported in part by a gift from the Renaissance Philanthropy Fund.

## Impact Statement

For certain architectures of deep neural networks, malicious users could efficiently sample randomness for the weights of the network in a way that enables them to generate colliding examples in the model $M$. These consist of pairs of fixed-precision inputs $(\mathbf{x}, \mathbf{x}')$ for which $\|M(\mathbf{x}) - M(\mathbf{x}')\|$ is unusually small.

For systems where such colliding examples provide significant utility (e.g., certain setups of facial recognition systems), an agent who has influence over the randomness used at the beginning of training has false positive or false negative capabilities that others do not. Our results are primarily theoretical. In particular, our work does not question or undermine the security or privacy of any system in use or near production.

Our work has a direct prevention strategy, namely, secure randomness generation. The history of cryptography has repeatedly seen instances in which control of randomness is surprisingly influential. We view our work as providing an example of this phenomenon in the context of deep neural networks instead of within standard cryptographic applications (e.g., encryption or authentication). As a result, this work does not require qualitatively new security solutions but rather points to the importance of secure randomness generation in ML.

If an honest party controls the randomness, we recommend NIST SP 800-90A/B/C as a concrete best practice for secure

random bit generation. Importantly, to prevent this risk, one should not delegate randomness generation for model weights to untrusted parties.

If one does not have access to a model's randomness and is simply using a model, an alternate mitigation strategy is to compel parties training a model to use randomness from a trusted randomness beacon (e.g., NIST randomness beacon). Here, comparing the weights of the model to the beacon's history is a simple verification approach that greatly limits adversarial behavior.

Stepping back, cryptography has an established tradition of openly documenting and studying potential vulnerabilities so that effective countermeasures can be taken well ahead of deployment. We believe our work naturally fits into this paradigm.

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

# A. Preliminaries

For a natural number $n \in \mathbb{N}$, we let $[n]$ denote the set $\{1, 2, \cdots, n\}$. For real numbers $a, b \in \mathbb{R}$ with $a \leq b$, we let $[a, b]$ denote the continuous interval $\{x \in \mathbb{R} : a \leq x \leq b\}$. Similarly, we let $(a, b)$ denote the open continuous interval $\{x \in \mathbb{R} : a < x < b\}$, and we let $[a, b)$ denote the continuous interval $\{x \in \mathbb{R} : a \leq x < b\}$. For $B \in \mathbb{N}$, we let $[-B : B]$ denote the discrete interval

$$[-B : B] = [-B, B] \cap \mathbb{Z} = \{-B, -B+1, \cdots, -1, 0, 1, \cdots, B-1, B\}.$$

We say a function $f : \mathbb{N} \to \mathbb{R}_{>0}$ is negligible if for all $c > 0$, $\lim_{n \to \infty} f(n) \cdot n^c = 0$. We use the notation $\mathrm{negl}(n)$ to denote a function that is negligible (in its input $n$). We similarly use the notation $\mathrm{poly}(n)$ to denote a function that is at most $n^{O(1)}$. As shorthand, we say an algorithm is p.p.t. if it runs in probabilistic polynomial time.

We let $\mathbb{1}(\varphi) \in \{0, 1\}$ denote the indicator variable corresponding to some logical predicate $\varphi$. For a set $S \subseteq \mathbb{R}$, we let $U(S)$ denote the uniform distribution over $S$, where the appropriate measure (i.e., discrete uniform or continuous uniform) will be clear from the choice of $S$. For a distribution $\mathcal{D}$ and $n \in \mathbb{N}$, we let $\mathcal{D}^n$ denote the distribution with $n$ i.i.d. samples from $\mathcal{D}$. We let $\mathcal{N}(\mu, \sigma^2)$ denote the univariate Gaussian (or normal) distribution with mean $\mu$ and variance $\sigma^2$. For a parameter $\gamma \in \mathbb{R}_{>0}$, we let $\mathcal{N}(\mu, \sigma^2)_{|\cdot| \leq \gamma}$ denote the conditional distribution of $X \sim \mathcal{N}(\mu, \sigma^2)$ given $|X| \leq \gamma$. For a vector $\boldsymbol{\mu} \in \mathbb{R}^n$ and a positive semi-definite matrix $\boldsymbol{\Sigma}$, we let $\mathcal{N}(\boldsymbol{\mu}, \boldsymbol{\Sigma})$ denote the multivariate Gaussian distribution with mean $\boldsymbol{\mu}$ and covariance matrix $\boldsymbol{\Sigma}$. Note that we allow $\boldsymbol{\Sigma}$ to be singular, in which case the multivariate Gaussian will be degenerate (i.e., have support in a proper subspace of $\mathbb{R}^n$). We let $\mathbf{I}_n \in \mathbb{R}^{n \times n}$ denote the identity matrix. We will use the fact that given $\boldsymbol{\mu}$ and $\boldsymbol{\Sigma}$, it is efficient to sample from $\mathcal{N}(\boldsymbol{\mu}, \boldsymbol{\Sigma})$, and similarly, given $\mu, \sigma$, and $\gamma$, it is efficient to sample from $\mathcal{N}(\mu, \sigma^2)_{|\cdot| \leq \gamma}$. For theoretical simplicity, we do not explicitly write out the finite precision of all computations, but all calculations will still go through with $\mathrm{poly}(n)$ bits of precision.

## A.1. Divergences

Let $\rho_0, \rho_1$ be density functions of distributions.

**Definition 1.** *The* Rényi divergence *between $\rho_1$ and $\rho_0$ is given by*

$$D_2(\rho_1 \| \rho_0) = \ln \left( \int \frac{\rho_1(x)^2}{\rho_0(x)} dx \right) = \ln \left( \mathop{\mathbb{E}}_{X \sim \rho_0} \left[ \frac{\rho_1(X)^2}{\rho_0(X)^2} \right] \right).$$

**Definition 2.** *The* Kullback-Leibler divergence *between $\rho_1$ and $\rho_0$ is given by*

$$d_{\mathrm{KL}}(\rho_1 \| \rho_0) = \int \rho_1(x) \ln \left( \frac{\rho_1(x)}{\rho_0(x)} \right) dx.$$

**Definition 3.** *The* total variation distance *between $\rho_1$ and $\rho_0$ is given by*

$$d_{\mathrm{TV}}(\rho_1, \rho_0) = \frac{1}{2} \int |\rho_1(x) - \rho_0(x)| \, dx.$$

**Lemma 1.** *For any two distributions $\rho_0$ and $\rho_1$,*

$$d_{\mathrm{TV}}(\rho_1, \rho_0) \leq \sqrt{\frac{d_{\mathrm{KL}}(\rho_1 \| \rho_0)}{2}} \leq \sqrt{\frac{D_2(\rho_1 \| \rho_0)}{2}}.$$

*Proof.* The left-hand inequality is Pinsker's inequality. The right-hand inequality is a standard fact of Rényi divergences (van Erven & Harremoës, 2014, Theorem 3). $\square$

**Lemma 2.** *For any density function $\rho_0$ and any nonnegative-valued function $f$, for the density function $\rho_1$ given by*

$$\rho_1(x) \propto \rho_0(x) f(x),$$

*it holds that*

$$D_2(\rho_1 \| \rho_0) = \ln \left( \frac{\mathbb{E}_{X \sim \rho_0} \left[ f(X)^2 \right]}{\mathbb{E}_{X \sim \rho_0} [f(X)]^2} \right).$$

*Proof.* For $\rho_1$ to be a normalized probability distribution, it must hold that

$$\rho_1(x) = \frac{\rho_0(x)f(x)}{\int \rho_0(x')f(x')dx'} = \frac{\rho_0(x)f(x)}{\mathbb{E}_{X\sim\rho_0}[f(X)]}.$$

We then have

$$\begin{aligned}
D_2(\rho_1\|\rho_0) &= \ln\left(\mathop{\mathbb{E}}_{X\sim\rho_0}\left[\frac{\rho_1(X)^2}{\rho_0(X)^2}\right]\right) \\
&= \ln\left(\mathop{\mathbb{E}}_{X\sim\rho_0}\left[\frac{\rho_0(X)^2 f(X)^2}{\mathbb{E}_{X'\sim\rho_0}[f(X')]^2 \rho_0(X)^2}\right]\right) \\
&= \ln\left(\mathop{\mathbb{E}}_{X\sim\rho_0}\left[\frac{f(X)^2}{\mathbb{E}_{X'\sim\rho_0}[f(X')]^2}\right]\right) \\
&= \ln\left(\frac{\mathbb{E}_{X\sim\rho_0}\left[f(X)^2\right]}{\mathbb{E}_{X\sim\rho_0}\left[f(X)\right]^2}\right),
\end{aligned}$$

as desired. $\square$

We now state the following standard fact of Rényi divergences.

**Lemma 3.** *For any two distributions $\rho_0$ and $\rho_1$ and any event $E$, we have*

$$\Pr_{\rho_0}(E) \geq \frac{\Pr_{\rho_1}(E)^2}{e^{D_2(\rho_1\|\rho_0)}}.$$

*Proof.* By Cauchy-Schwarz, we have

$$\begin{aligned}
\Pr_{\rho_1}(E) = \mathop{\mathbb{E}}_{X\sim\rho_1}[\mathbb{1}(X\in E)] &= \mathop{\mathbb{E}}_{X\sim\rho_0}\left[\mathbb{1}(X\in E)\cdot\frac{\rho_1(X)}{\rho_0(X)}\right] \\
&\leq \sqrt{\mathop{\mathbb{E}}_{X\sim\rho_0}[\mathbb{1}(X\in E)^2]\cdot\mathop{\mathbb{E}}_{X\sim\rho_0}\left[\frac{\rho_1(X)^2}{\rho_0(X)^2}\right]} \\
&= \sqrt{\Pr_{\rho_0}(E)\cdot e^{D_2(\rho_1\|\rho_0)}}.
\end{aligned}$$

Rearranging gives the desired result. $\square$

## A.2. Number Balancing and Symmetric Binary Perceptrons

We define the number balancing problem.

**Definition 4.** *The* number balancing problem (NBP) *with parameters $\kappa : \mathbb{N} \to \mathbb{R}_{>0}$ and $B : \mathbb{N} \to \mathbb{N}$ is defined as follows. On input $\mathbf{a} \sim \mathcal{N}(0,1)^n$, output $\mathbf{x} \in [-B : B]^n \setminus \{0^n\}$ such that $|\langle \mathbf{a}, \mathbf{x}\rangle| \leq \kappa\sqrt{n}$, where $\kappa = \kappa(n)$ and $B = B(n)$. If unspecified, we take $B(n) = 1$.*

For $\kappa(n) \geq \Theta(1/2^n)$, we know that there exist $\{\pm 1\}^n$ solutions to NBP with high probability (so, in particular, there exist $[-B : B]^n \setminus \{0^n\}$ solutions) (Karmarkar et al., 1986). The best polynomial time algorithm, due to Karmarkar and Karp, achieves $\kappa(n) = 1/2^{\Theta(\log^2 n)}$ (Karmarkar & Karp, 1982) (for the most stringent case of $B = 1$).

For $\kappa(n) \leq 1/2^{\log^{3+\varepsilon} n}$, we have computational hardness assuming sub-exponential hardness of worst-case lattice problems (Vafa & Vaikuntanathan, 2025). Therefore, the following assumption is true assuming worst-case lattice problems are hard to solve:

**Assumption 1.** *For all p.p.t. algorithms $\mathcal{A}$ and $\varepsilon > 0$, and $B \leq \mathrm{poly}(n)$,*

$$\Pr_{\mathbf{a}\sim\mathcal{N}(0,1)^n}\left(\mathbf{x} \leftarrow \mathcal{A}(\mathbf{a}) : \mathbf{x} \in [-B : B]^n \setminus \{0^n\} \ \wedge \ |\langle \mathbf{a}, \mathbf{x}\rangle| \leq \frac{1}{2^{\log(n)^{3+\varepsilon}}}\right) = \mathrm{negl}(n).$$

We can similarly define the symmetric binary perceptron problem.

**Definition 5.** *The* symmetric bounded perceptron (SBP) *problem with parameters* $\kappa : \mathbb{N} \to \mathbb{R}_{>0}$, $m : \mathbb{N} \to \mathbb{N}$, *and* $B : \mathbb{N} \to \mathbb{N}$ *is defined as follows. On input* $\mathbf{A} \sim \mathcal{N}(0,1)^{m \times n}$, *output* $\mathbf{x} \in [-B : B]^n \setminus \{0^n\}$ *such that* $\|\mathbf{A}\mathbf{x}\|_\infty \leq \kappa \sqrt{n}$, *where* $\kappa = \kappa(n)$, $m = m(n)$, *and* $B = B(n)$. *If unspecified, we take* $B(n) = 1$.

For $\kappa \geq \Theta(2^{-n/m})$, we know that there exist $\{\pm 1\}^n$ solutions to SBP with high probability (so, in particular, there exist $[-B : B]^n \setminus \{0^n\}$ solutions) (Aubin et al., 2019; Perkins & Xu, 2021; Abbe et al., 2021). The best polynomial time algorithm, due to Bansal and Spencer (Bansal, 2010; Bansal & Spencer, 2020), achieves $\kappa = O\left(\sqrt{m/n}\right)$ (for the most stringent case of $B = 1$).

For $B, n \leq \mathrm{poly}(m)$ and $\kappa \leq 1/(\sqrt{n} \cdot m^\varepsilon)$, we have computational hardness assuming polynomial hardness of worst-case lattice problems (Vafa & Vaikuntanathan, 2025; Bogdanov et al., 2025). Therefore, the following assumption is true assuming worst-case lattice problems are hard to solve:

**Assumption 2.** *For all p.p.t. algorithms* $\mathcal{A}$, $\varepsilon > 0$, *and* $B, n \leq \mathrm{poly}(m)$,

$$\Pr_{\mathbf{A} \sim \mathcal{N}(0,1)^{m \times n}} \left( \mathbf{x} \leftarrow \mathcal{A}(\mathbf{A}) : \mathbf{x} \in [-B : B]^n \setminus \{0^n\} \ \wedge \ \|\mathbf{A}\mathbf{x}\|_\infty \leq \frac{1}{m^\varepsilon} \right) = \mathrm{negl}(n).$$

## B. Backdoors for Random Gaussian Projections

The goal of this section is to prove the following theorem.

**Theorem 2.** *For all* $m \leq n$, *there is a p.p.t. algorithm* $\mathrm{BackdoorMatrix}(1^n, 1^m)$ *that outputs a matrix* $\mathbf{A} \in \mathbb{R}^{m \times n}$ *and a vector* $\mathbf{z} \in \{\pm 1\}^n$ *such that the following hold:*

- *We have*

$$\|\mathbf{A}\mathbf{z}\|_\infty \leq O\left(\frac{\sqrt{n}}{2^{n/m}}\right).$$

- *We have the statistical bounds*

$$d_{\mathrm{TV}}\left(\mathbf{A}, \mathcal{N}(0,1)^{m \times n}\right) = O\left(\sqrt{\frac{m}{n} \log(m/n) + e^{-\Omega(m)}}\right),$$

$$D_2\left(\mathbf{A} \| \mathcal{N}(0,1)^{m \times n}\right) = O\left(\frac{m}{n} \log(m/n) + e^{-\Omega(m)}\right).$$

- *The marginal distribution of* $\mathbf{z}$ *is uniform over* $\{\pm 1\}^n$.

Note that if $m = \omega(1)$ and $m = o(n)$, both statistical divergences become $o(1)$.

We also give a version of this theorem with slightly different parameters in the regime where $m = \Theta(1)$ (i.e., $m$ is fixed while $n$ grows).

**Theorem 3.** *For all* $m = \Theta(1)$ *and growing* $n$, *there is a universal constant* $C > 0$ *and a p.p.t. algorithm* $\mathrm{BackdoorMatrix}(1^n, 1^m)$ *that outputs a matrix* $\mathbf{A} \in \mathbb{R}^{m \times n}$ *and a vector* $\mathbf{z} \in \{\pm 1\}^n$ *such that the following hold:*

- *We have*

$$\|\mathbf{A}\mathbf{z}\|_\infty \leq O\left(\frac{n^C}{2^{n/m}}\right).$$

- *We have the statistical distance bounds*

$$d_{\mathrm{TV}}\left(\mathbf{A}, \mathcal{N}(0,1)^{m \times n}\right) = O\left(\sqrt{\frac{\log n}{n}}\right),$$

$$D_2\left(\mathbf{A} \| \mathcal{N}(0,1)^{m \times n}\right) = O\left(\frac{\log n}{n}\right).$$

- *The marginal distribution of* $\mathbf{z}$ *is uniform over* $\{\pm 1\}^n$.

**B.1. Sampling the Backdoor**

---



Matrix Backdoor Construction

$\mathrm{BackdoorMatrix}(1^n, 1^m)$:

1. Sample $\mathbf{z} \sim U(\{\pm 1\}^n)$.

2. For $i \in [m]$:

    (a) Sample $b_i \sim \mathcal{N}(0, n)_{|\cdot| \leq \kappa\sqrt{n}}$.

    (b) Sample vector $\mathbf{a}_i \sim \mathcal{N}\left(\frac{b_i}{n} \cdot \mathbf{z}, \mathbf{I}_n - \frac{1}{n}\mathbf{z}\mathbf{z}^\top\right) = \mathcal{N}\left(\mathbf{0}, \mathbf{I}_n \mid \mathbf{a}_i^\top \mathbf{z} = b_i\right)$.

3. Define $\mathbf{A} \in \mathbb{R}^{m \times n}$ to have rows $\mathbf{a}_1, \cdots, \mathbf{a}_m \in \mathbb{R}^n$.

4. Output $(\mathbf{A}, \mathbf{z})$.



*Figure 4.* Description of the matrix backdoor algorithm used in Theorems 2 and 3.

Define $\mu_0$ to be the joint distribution defined implicitly via the following process:

1. Sample $\mathbf{A} \sim \mathcal{N}(0, 1)^{m \times n}$.

2. Sample $\mathbf{z} \sim U(\{\pm 1\}^n)$.

3. Set $\mathbf{b} = \mathbf{A}\mathbf{z} \in \mathbb{R}^m$.

4. Output $(\mathbf{A}, \mathbf{z}, \mathbf{b}) \in \mathbb{R}^{m \times n} \times \{\pm 1\}^n \times \mathbb{R}^m$.

More explicitly, the density is given by

$$\mu_0(\mathbf{A}, \mathbf{z}, \mathbf{b}) = \frac{1}{(2\pi)^{mn/2}} e^{-\frac{1}{2}\sum_{i,j} A_{i,j}^2} \cdot \frac{1}{2^n} \cdot \delta(\mathbf{b} - \mathbf{A}\mathbf{z}),$$

where $\delta()$ is the delta function generalized to $\mathbb{R}^m$, i.e.,

$$\int_{\mathbb{R}^m} \delta(\mathbf{y}) f(\mathbf{y}) d\mathbf{y} = f(\mathbf{0}).$$

Now, define the distribution $\mu_1$ to be the distribution $\mu_0$ conditioned on $\|\mathbf{b}\|_\infty \leq \kappa\sqrt{n}$. That is,

$$\mu_1(\mathbf{A}, \mathbf{z}, \mathbf{b}) \propto \frac{1}{(2\pi)^{mn/2}} e^{-\frac{1}{2}\sum_{i,j} A_{i,j}^2} \cdot \frac{1}{2^n} \cdot \delta(\mathbf{b} - \mathbf{A}\mathbf{z}) \cdot \mathbb{1}\left(\|\mathbf{b}\|_\infty \leq \kappa\sqrt{n}\right)$$

$$\propto e^{-\frac{1}{2}\sum_{i,j} A_{i,j}^2} \cdot \delta(\mathbf{b} - \mathbf{A}\mathbf{z}) \cdot \mathbb{1}\left(\|\mathbf{b}\|_\infty \leq \kappa\sqrt{n}\right).$$

Let $\rho_0$ and $\rho_1$ denote the marginal distributions on $\mathbf{A}$ in $\mu_0$ and $\mu_1$, respectively. Note that $\rho_0$ is identically $\mathcal{N}(0, 1)^{m \times n}$. Here, we relate $\rho_1$ and the algorithm $\mathrm{BackdoorMatrix}$ given in Figure 4.

**Claim 1.** *The output distribution of $\mathbf{A}$ in* $\mathrm{BackdoorMatrix}$ *(as given in Figure 4) is identical to $\rho_1$.*

*Proof.* For any fixed $\mathbf{z} \in \{\pm 1\}^n$, the distribution of $\mathbf{b} = \mathbf{A}\mathbf{z}$ is $\mathcal{N}(0, \|\mathbf{z}\|_2^2)^m = \mathcal{N}(0, n)^m$ over random $\mathbf{A} \sim \mathcal{N}(0, 1)^{m \times n}$. In particular, in $\mu_0$, $\mathbf{z}$ and $\mathbf{b}$ are independent. Therefore, $\mu_0$ can be identically described as follows, by first conditioning on $\mathbf{z}$ and then on $\mathbf{z}$ and $\mathbf{b}$ together:

1. Sample $\mathbf{z} \sim U(\{\pm 1\}^n)$.

2. Sample $\mathbf{b} \sim \mathcal{N}(0, n)^m$.

3. Sample $\mathbf{a}_1, \cdots, \mathbf{a}_m \sim \mathcal{N}(0,1)^n$ conditioned on $b_i = \mathbf{a}_i^\top \mathbf{z}$ for all $i \in [m]$. Let $\mathbf{A}$ be the matrix that has rows given by $\mathbf{a}_i$.

4. Output $(\mathbf{A}, \mathbf{z}, \mathbf{b})$.

In this formulation, we can describe $\mu_1$ as follows, where all we change from the above is that we condition on $\|\mathbf{b}\|_\infty$.

1. Sample $\mathbf{z} \sim U(\{\pm 1\}^n)$.

2. Sample $b_1, \cdots, b_m \sim \mathcal{N}(0,n)_{|\cdot| \le \kappa\sqrt{n}}$, and let $\mathbf{b} = (b_1, \cdots, b_m) \in \mathbb{R}^m$.

3. Sample $\mathbf{a}_1, \cdots, \mathbf{a}_m \sim \mathcal{N}(0,1)^n$ conditioned on $b_i = \mathbf{a}_i^\top \mathbf{z}$ for all $i \in [m]$. Let $\mathbf{A}$ be the matrix that has rows given by $\mathbf{a}_i$.

4. Output $(\mathbf{A}, \mathbf{z}, \mathbf{b})$.

More explicitly, sampling $\mathbf{a}_i \sim \mathcal{N}(0,1)^n$ conditioned on $b_i = \mathbf{a}_i^\top = \mathbf{z}$ is equivalent to sampling

$$\mathbf{a}_i \sim \mathcal{N}\left(0, \mathbf{I}_n \mid \mathbf{a}_i^\top \mathbf{z} = b_i\right) = \mathcal{N}\left(\frac{b_i}{n} \cdot \mathbf{z}, \mathbf{I}_n - \frac{1}{n}\mathbf{z}\mathbf{z}^\top\right).$$

This description of $\mu_1$ is now exactly the one given in Figure 4. The claim follows. □

Let $N : \mathbb{R}^{m \times n} \to \mathbb{N}$ denote the function

$$N(\mathbf{A}) = \left|\{\mathbf{z} \in \{\pm 1\}^n : \|\mathbf{A}\mathbf{z}\|_\infty \le \kappa\sqrt{n}\}\right| = \sum_{\mathbf{z} \in \{\pm 1\}^n} \mathbb{1}\left(\|\mathbf{A}\mathbf{z}\|_\infty \le \kappa\sqrt{n}\right). \tag{2}$$

**Claim 2.** *We have*

$$\rho_1(\mathbf{A}) \propto \rho_0(\mathbf{A}) \cdot N(\mathbf{A}).$$

*Proof.* By marginalizing out over $\mathbf{z}$ and $\mathbf{b}$, we have

$$\rho_1(\mathbf{A}) = \sum_{\mathbf{z} \in \{\pm 1\}^n} \int_{\mathbb{R}^m} \mu_1(\mathbf{A}, \mathbf{z}, \mathbf{b}) \cdot d\mathbf{b}$$

$$\propto \sum_{\mathbf{z} \in \{\pm 1\}^n} \int_{\mathbb{R}^m} e^{-\frac{1}{2}\sum_{i,j} A_{i,j}^2} \cdot \delta(\mathbf{b} - \mathbf{A}\mathbf{z}) \cdot \mathbb{1}\left(\|\mathbf{b}\|_\infty \le \kappa\sqrt{n}\right) \cdot d\mathbf{b}$$

$$= \sum_{\mathbf{z} \in \{\pm 1\}^n} \int_{\left[-\kappa\sqrt{n}, \kappa\sqrt{n}\right]^m} e^{-\frac{1}{2}\sum_{i,j} A_{i,j}^2} \cdot \delta(\mathbf{b} - \mathbf{A}\mathbf{z}) \cdot d\mathbf{b}$$

$$= e^{-\frac{1}{2}\sum_{i,j} A_{i,j}^2} \sum_{\mathbf{z} \in \{\pm 1\}^n} \int_{\left[-\kappa\sqrt{n}, \kappa\sqrt{n}\right]^m} \delta(\mathbf{b} - \mathbf{A}\mathbf{z}) \cdot d\mathbf{b}$$

$$= e^{-\frac{1}{2}\sum_{i,j} A_{i,j}^2} \sum_{\mathbf{z} \in \{\pm 1\}^n} \mathbb{1}\left(\|\mathbf{A}\mathbf{z}\|_\infty \le \kappa\sqrt{n}\right)$$

$$= e^{-\frac{1}{2}\sum_{i,j} A_{i,j}^2} \cdot N(\mathbf{A})$$

$$\propto \rho_0(\mathbf{A}) \cdot N(\mathbf{A}),$$

as desired. □

**Claim 3.** *For* $\mathbf{A}$ *output by* BackdoorMatrix*, we have*

$$D_2\left(\mathbf{A} \| \mathcal{N}(0,1)^{m \times n}\right) = \ln\left(\frac{\mathbb{E}_{\mathbf{A} \sim \mathcal{N}(0,1)^{m \times n}}\left[N(\mathbf{A})^2\right]}{\mathbb{E}_{\mathbf{A} \sim \mathcal{N}(0,1)^{m \times n}}\left[N(\mathbf{A})\right]^2}\right).$$

*Proof.* This directly follows by combining Claim 1, Claim 2, and Lemma 2. □

### B.2. Concentration in the Number of Solutions

As in (2), let $N = N(\mathbf{A})$ denote the number of $\pm 1$ solutions $\mathbf{z}$ to $\|\mathbf{A}\mathbf{z}\|_\infty \le \kappa\sqrt{n}$ for $\mathbf{A} \sim \mathcal{N}(0,1)^{m \times n}$, and let $\alpha = m/n$. Let $\phi(\kappa) = \Pr(|Z| \le \kappa)$ for a standard normal $Z \sim \mathcal{N}(0,1)$. For small $\kappa$, $\sqrt{\pi/2} \cdot \phi(\kappa) \approx \kappa$. More precisely,

$$\kappa - \frac{\kappa^3}{6} \le \sqrt{\frac{\pi}{2}} \cdot \phi(\kappa) \le \kappa.$$

**Proposition 1.** *Assuming* $\phi(\kappa) \ge 2^{-(1-\epsilon)/\alpha}$,

$$\frac{\mathbb{E}\left[N^2\right]}{\mathbb{E}\left[N\right]^2} \le \frac{1}{\sqrt{1 - \alpha\lambda(\epsilon)}} + 2\exp{-\Omega(\epsilon n)}$$

*whenever* $\alpha\lambda(\epsilon) < 1$, *where* $\lambda(\epsilon) = O(\log 1/\epsilon)$.

In the special case $m = 1$, (Karmarkar et al., 1986) calculated the tight bound $1 + \pi n/\kappa 2^n \pm O(1/n)$ on the moment ratio for the count of perfectly balanced solutions only. In the extreme regime $\kappa \approx n^{O(1)}2^{-n}$ our bound is worse by a factor logarithmic in $n$. We did not attempt to remove this factor. In the regime of constant $m$ and increasing $n$ Dyer and Frieze (Dyer & Frieze, 1989) give an asymptotic upper bound of $1 + o(1)$ without specifying the lower-order dependence. Their calculations are substantially more complicated as they pertain to values of $\kappa$ very close to the statistical threshold (below which $N$ is very likely to be zero).

**Corollary 1.** *There exist universal constants* $C_1, C_2 > 0$ *such that for all* $m = o(n)$ *and* $\kappa = C_1 \cdot 2^{-n/m}$, *it holds that*

$$\frac{\mathbb{E}\left[N^2\right]}{\mathbb{E}\left[N\right]^2} \le 1 + O\left(\frac{m}{n} \cdot \log(n/m) + e^{-C_2 m}\right).$$

*In particular, if it additionally holds that* $m = \omega(1)$, *we have*

$$\frac{\mathbb{E}\left[N^2\right]}{\mathbb{E}\left[N\right]^2} \le 1 + o(1).$$

*Proof.* Let $\alpha = m/n = o(1)$. Set $\epsilon = \Theta(\alpha) = o(1)$ in Proposition 1 (in terms of $C_1$) so that for $\kappa = C_1 \cdot 2^{-n/m}$, it holds that $\phi(\kappa) \ge 2^{-(1-\epsilon)n/m}$. As $\lambda(\epsilon) \le O(\log(1/\epsilon)) \le O(\log(n/m))$, we have

$$\alpha\lambda(\epsilon) \le O(\alpha \log(1/\alpha)) = o(1).$$

In particular, $\alpha\lambda(\epsilon) < 1$ and $1/\sqrt{1 - \alpha\lambda(\epsilon)} < 1 + O(\alpha\lambda(\epsilon))$ for sufficiently small $\alpha$. Therefore, by Proposition 1, we have

$$\frac{\mathbb{E}\left[N^2\right]}{\mathbb{E}\left[N\right]^2} \le 1 + O(\alpha\lambda(\epsilon)) + 2e^{-\Omega(\epsilon n)} \le 1 + O(\alpha \log(1/\alpha)) + 2e^{-\Omega(m)},$$

as desired. $\square$

We now give a slightly different parameter setting that gives a $1 + o(1)$ bound for any $m = O(1)$.

**Corollary 2.** *There exists a universal constant* $C_1 > 0$ *such that for all* $m = o(n)$ *and* $\kappa = n^{C_1} \cdot 2^{-n/m}$, *it holds that*

$$\frac{\mathbb{E}\left[N^2\right]}{\mathbb{E}\left[N\right]^2} \le 1 + O\left(\frac{m}{n} \cdot \log(n/m) + e^{-2m \log n}\right).$$

*In particular, for* $m = \Theta(1)$ *and growing* $n$, *we have*

$$\frac{\mathbb{E}\left[N^2\right]}{\mathbb{E}\left[N\right]^2} \le 1 + O\left(\frac{\log n}{n}\right).$$

*Proof.* Let $\alpha = m/n = o(1)$. Set $\epsilon = C_2 \alpha \log_2 n$ and $C_2$ in terms of $C_1$ so that for $\kappa = n^{C_1} \cdot 2^{-n/m}$, we have $\phi(\kappa) \geq 2^{-(1-\epsilon)/\alpha} = n^{C_2} \cdot 2^{-n/m}$. As $\lambda(\epsilon) \leq O(\log(1/\epsilon)) \leq O(\log(1/\alpha))$, we have $\alpha\lambda(\epsilon) = o(1)$, which in particular means $1/\sqrt{1 - \alpha\lambda(\epsilon)} < 1 + O(\alpha\lambda(\epsilon))$ for sufficiently small $\alpha$. Therefore, by Proposition 1, setting $C_1$ sufficiently large, we have

$$\frac{\mathbb{E}\left[N^2\right]}{\mathbb{E}\left[N\right]^2} \leq 1 + O(\alpha\lambda(\epsilon)) + 2e^{-\Omega(\epsilon n)} \leq 1 + O(\alpha \log(1/\alpha)) + 2e^{-2m \log n},$$

as desired. $\qquad\square$

**Proof of Proposition 1**    We first show the following claim.

**Claim 4.** *Let $\rho$ be the position of an $n$-step $\pm 1$ random walk divided by $n$. Then*

$$\frac{\mathbb{E}\left[N^2\right]}{\mathbb{E}\left[N\right]^2} = \mathbb{E}_{\rho}\left[\left(\frac{\Pr(|Z'| \leq \kappa \mid |Z| \leq \kappa)}{\Pr(|Z| \leq \kappa)}\right)^m\right], \tag{3}$$

*where $Z, Z'$ are $\rho$-correlated standard normal, i.e.,*

$$(Z, Z') \sim \mathcal{N}\left(\begin{pmatrix} 0 \\ 0 \end{pmatrix}, \begin{pmatrix} 1 & \rho \\ \rho & 1 \end{pmatrix}\right).$$

*Proof of Claim 4.*  Let

$$q = \phi(\kappa) = \Pr_{Z \sim \mathcal{N}(0,1)}(|Z| \leq \kappa) = \Pr_{\mathbf{a} \sim \mathcal{N}(0,1)^n}\left(\left|\mathbf{a}^\top \mathbf{x}\right| \leq \kappa\sqrt{n}\right),$$

where $\mathbf{x} \in \mathbb{R}^n$ is any fixed vector with $\|\mathbf{x}\|_2 = \sqrt{n}$. By linearity of expectation and definition of $N = N(\mathbf{A})$, it follows that

$$\mathbb{E}[N] = \sum_{\mathbf{x} \in \{\pm 1\}^n} \Pr_{\mathbf{A} \sim \mathcal{N}(0,1)^{m \times n}}\left(\|\mathbf{A}\mathbf{x}\|_\infty \leq \kappa\sqrt{n}\right)$$

$$= \sum_{\mathbf{x} \in \{\pm 1\}^n} \left(\Pr_{\mathbf{a} \sim \mathcal{N}(0,1)^n}\left(\left|\mathbf{a}^\top \mathbf{x}\right| \leq \kappa\sqrt{n}\right)\right)^m = 2^n q^m.$$

For the second moment, we have

$$\mathbb{E}\left[N^2\right] = \sum_{\mathbf{x}_1, \mathbf{x}_2 \in \{\pm 1\}^n} \Pr_{\mathbf{A} \sim \mathcal{N}(0,1)^{m \times n}}\left(\|\mathbf{A}\mathbf{x}_1\|_\infty \leq \kappa\sqrt{n}, \|\mathbf{A}\mathbf{x}_2\|_\infty \leq \kappa\sqrt{n}\right)$$

$$= \sum_{\mathbf{x}_1, \mathbf{x}_2 \in \{\pm 1\}^n} \Pr_{\mathbf{a} \sim \mathcal{N}(0,1)^n}\left(\left|\mathbf{a}^\top \mathbf{x}_1\right| \leq \kappa\sqrt{n}, \left|\mathbf{a}^\top \mathbf{x}_2\right| \leq \kappa\sqrt{n}\right)^m.$$

A quick calculation reveals that for $\mathbf{a} \sim \mathcal{N}(0,1)^n$ and $\mathbf{x}_1, \mathbf{x}_2 \in \{\pm 1\}^n$, we have

$$\left(\mathbf{a}^\top \mathbf{x}_1, \mathbf{a}^\top \mathbf{x}_2\right) \sim \mathcal{N}\left(\begin{pmatrix} 0 \\ 0 \end{pmatrix}, \begin{pmatrix} n & n - 2 \cdot \Delta(\mathbf{x}_1, \mathbf{x}_2) \\ n - 2 \cdot \Delta(\mathbf{x}_1, \mathbf{x}_2) & n \end{pmatrix}\right),$$

where $\Delta(\mathbf{x}_1, \mathbf{x}_2)$ is the Hamming distance between $\mathbf{x}_1$ and $\mathbf{x}_2$ (i.e., counts the number of distinct coordinates). By rescaling, we can write

$$\mathbb{E}\left[N^2\right] = \sum_{\mathbf{x}_1, \mathbf{x}_2 \in \{\pm 1\}^n} \Pr_{\mathbf{a} \sim \mathcal{N}(0,1)^n}\left(\left|\mathbf{a}^\top \mathbf{x}_1\right| \leq \kappa\sqrt{n}, \left|\mathbf{a}^\top \mathbf{x}_2\right| \leq \kappa\sqrt{n}\right)^m$$

$$= \sum_{k=0}^n \sum_{\substack{\mathbf{x}_1, \mathbf{x}_2 \\ \Delta(\mathbf{x}_1, \mathbf{x}_2) = k}} \Pr_{Z_1, Z_2 \ (1 - 2k/n)\text{-corr.}}(|Z_1| \leq \kappa, |Z_2| \leq \kappa)^m$$

$$= 2^n \sum_{k=0}^n \binom{n}{k} \Pr_{Z_1, Z_2 \ (1 - 2k/n)\text{-corr.}}(|Z_1| \leq \kappa, |Z_2| \leq \kappa)^m$$

$$= 2^{2n} \mathbb{E}_{\rho} \Pr_{Z_1, Z_2 \ \rho\text{-corr.}}(|Z_1| \leq \kappa, |Z_2| \leq \kappa)^m$$

$$= 2^{2n} q^m \mathbb{E}_{\rho} \Pr_{Z_1, Z_2 \ \rho\text{-corr.}}(|Z_2| \leq \kappa \mid |Z_1| \leq \kappa)^m,$$

where $\rho$ is the position of an $n$-step $\pm 1$ random walk divided by $n$.

We can combine the first and second moment calculations to get

$$
\frac{\mathbb{E}\left[N^2\right]}{\mathbb{E}\left[N\right]^2} = \frac{2^{2n}q^m}{2^{2n}q^{2m}} \cdot \mathop{\mathbb{E}}_{\rho} \mathop{\Pr}_{Z_1, Z_2 \ \rho\text{-corr.}} \left(|Z_2| \leq \kappa \mid |Z_1| \leq \kappa\right)^m
$$

$$
= \mathop{\mathbb{E}}_{\rho} \left[\left(\frac{\Pr_{Z_1, Z_2 \ \rho\text{-corr.}} \left(|Z_2| \leq \kappa \mid |Z_1| \leq \kappa\right)}{q}\right)^m\right],
$$

as desired. □

Since $Z'$ can be written as $\rho Z + \sqrt{1 - \rho^2} Y$ for some independent $Y \sim \mathcal{N}(0, 1)$, and among all fixed variance Gaussians the measure of an interval is maximized by the one that is centered, the numerator of the quantity in Claim 4 can be upper bounded by

$$
\Pr\left(\left|\sqrt{1 - \rho^2} \cdot Y\right| \leq \kappa\right) = \Pr\left(|Y| \leq \frac{\kappa}{\sqrt{1 - \rho^2}}\right) \leq \frac{\Pr(|Y| \leq \kappa)}{\sqrt{1 - \rho^2}}.
$$

(The inequality can be verified by a change of variables in the Gaussian integral.) Therefore,

$$
\frac{\Pr(|Z'| \leq \kappa \mid |Z| \leq \kappa)}{\Pr(|Z| \leq \kappa)} \leq \frac{1}{\sqrt{1 - \rho^2}}.
$$

As the ratio is also at most $1/\Pr(|Z| \leq \kappa)$, for every $\delta > 0$ we obtain as a consequence of Claim 4 that

$$
\frac{\mathbb{E}\left[N^2\right]}{\mathbb{E}\left[N\right]^2} \leq \mathbb{E}\left[\frac{1}{(1 - \rho^2)^{m/2}} \cdot \mathbb{1}(|\rho| < 1 - \delta)\right] + \frac{\Pr(|\rho| \geq 1 - \delta)}{\phi(\kappa)^m}. \tag{4}
$$

By standard tail bounds on the binomial distribution, we have

$$
\Pr(|\rho| \geq 1 - \delta) \leq 2 \cdot 2^{n(H(\delta/2) - 1)},
$$

where $H$ denotes the binary entropy function.

Therefore, the second term in (4) is at most

$$
\frac{2 \cdot 2^{n(H(\delta/2) - 1)}}{\phi(\kappa)^m} = 2 \cdot 2^{(\alpha \log(1/\phi(\kappa)) - 1 + H(\delta/2))n},
$$

Choosing $\delta < 1$ so that $H(\delta/2) = \epsilon/2$ makes this at most $2 \exp(-\Omega(\epsilon n))$ under our assumption on $\kappa$.

For the first term in (4), we use the next bound which follows from the convexity of $\exp$.

**Fact 1.** *For $|\rho| < 1 - \delta$, we have $1 - \rho^2 > \exp(-\lambda \rho^2)$, where $\lambda = -\ln(2\delta - \delta^2)/(1 - \delta)^2$.*

Therefore,

$$
\mathbb{E}\left[\frac{1}{(1 - \rho^2)^{m/2}} \cdot \mathbb{1}(|\rho| < 1 - \delta)\right] \leq \mathbb{E}\left[\exp\left(\lambda \rho^2 m/2\right) \cdot \mathbb{1}(|\rho| < 1 - \delta)\right]
$$

$$
\leq \mathbb{E}\left[\exp\left(\lambda \rho^2 m/2\right)\right].
$$

**Claim 5.** $\mathbb{E}\left[\exp\left(t\rho^2 n\right)\right] \leq \mathbb{E}\left[\exp\left(tZ^2\right)\right]$ *where $t \geq 0$ and $Z$ is a standard normal.*

*Proof.* It suffices to show that the even moments of $\rho\sqrt{n}$ are dominated by those of $Z$. Both $\rho\sqrt{n}$ and $Z$ have the form $(X_1 + \cdots + X_n)/\sqrt{n}$, where the $X_i$ are i.i.d. Rademacher and standard normal, respectively. As the Rademacher moments are dominated by the standard normal ones, the same must be true for $\rho\sqrt{n}$ and $Z$. □

The squared normal moment generating function $\mathbb{E}\left[\exp\left(tZ^2\right)\right]$ evaluates to $1/\sqrt{1-2t}$ when $t < 1/2$ (and is unbounded otherwise) so, by plugging in $t = \lambda\alpha/2 = \lambda m/(2n)$,

$$\mathbb{E}\left[\frac{1}{(1-\rho^2)^{m/2}} \cdot \mathbb{1}(|\rho| < 1-\delta)\right] \leq \mathbb{E}\left[\exp\left(\lambda\rho^2 m/2\right)\right] \leq \mathbb{E}\left[\exp\left(\lambda\alpha Z^2/2\right)\right] = \frac{1}{\sqrt{1-\lambda\alpha}},$$

provided $\lambda < 1/\alpha$. For small $\epsilon$, by using standard bounds on the binary entropy function $H$, we have

$$\lambda = O(\log(O(1/\delta))) = O(\log(O(1/H^{-1}(\epsilon/2)))) = O(\log(1/\epsilon)),$$

as desired.

### B.3. Putting It All Together

*Proof of Theorem 2.* Consider the algorithm $\mathrm{BackdoorMatrix}(1^n, 1^m)$ given in Figure 4 where $\kappa = O(2^{-n/m})$. By construction, for all $i \in [m]$,

$$\left|\mathbf{a}_i^\top \mathbf{z}\right| = |b_i| \leq \kappa\sqrt{n},$$

so we have

$$\|\mathbf{A}\mathbf{z}\|_\infty = \max_{i\in[m]}\left|\mathbf{a}_i^\top \mathbf{z}\right| \leq \kappa\sqrt{n} \leq O\left(\sqrt{n} \cdot 2^{-n/m}\right).$$

By Claim 3, we have

$$D_2\left(\mathbf{A}\|\mathcal{N}(0,1)^{m\times n}\right) = \ln\left(\frac{\mathbb{E}_{\mathbf{A}\sim\mathcal{N}(0,1)^{m\times n}}\left[N(\mathbf{A})^2\right]}{\mathbb{E}_{\mathbf{A}\sim\mathcal{N}(0,1)^{m\times n}}\left[N(\mathbf{A})\right]^2}\right).$$

By Corollary 1 and choosing the constant in $\kappa = O(2^{-n/m})$ appropriately, we have

$$\frac{\mathbb{E}_{\mathbf{A}\sim\mathcal{N}(0,1)^{m\times n}}\left[N(\mathbf{A})^2\right]}{\mathbb{E}_{\mathbf{A}\sim\mathcal{N}(0,1)^{m\times n}}\left[N(\mathbf{A})\right]^2} \leq 1 + O\left(\frac{m}{n} \cdot \log(n/m) + e^{-\Omega(m)}\right).$$

Therefore, by Lemma 1 and the inequality $\ln(1+x) \leq x$,

$$\begin{aligned}
d_{\mathrm{TV}}\left(\mathbf{A}, \mathcal{N}(0,1)^{m\times n}\right) &\leq O\left(\sqrt{D_2\left(\mathbf{A}\|\mathcal{N}(0,1)^{m\times n}\right)}\right) \\
&= O\left(\sqrt{\ln\left(\frac{\mathbb{E}_{\mathbf{A}\sim\mathcal{N}(0,1)^{m\times n}}\left[N(\mathbf{A})^2\right]}{\mathbb{E}_{\mathbf{A}\sim\mathcal{N}(0,1)^{m\times n}}\left[N(\mathbf{A})\right]^2}\right)}\right) \\
&\leq O\left(\sqrt{\ln\left(1 + O\left(\frac{m}{n} \cdot \log(n/m) + e^{-\Omega(m)}\right)\right)}\right) \\
&\leq O\left(\sqrt{\frac{m}{n} \cdot \log(n/m) + e^{-\Omega(m)}}\right),
\end{aligned}$$

as desired.

Finally, it is clear from inspection of $\mathrm{BackdoorMatrix}$ in Figure 4 that the marginal distribution on $\mathbf{z}$ is uniform over $\{\pm 1\}^n$. $\qquad\square$

*Proof of Theorem 3.* The proof is exactly like that of Theorem 2, with the only difference being the bound for the concentration in the number of solutions. For $\kappa = n^C 2^{-n/m}$ for appropriately chosen constant $C$, by Corollary 2, we have

$$\frac{\mathbb{E}_{\mathbf{A}\sim\mathcal{N}(0,1)^{m\times n}}\left[N(\mathbf{A})^2\right]}{\mathbb{E}_{\mathbf{A}\sim\mathcal{N}(0,1)^{m\times n}}\left[N(\mathbf{A})\right]^2} \leq 1 + O\left(\frac{\log n}{n}\right).$$

Therefore, by Lemma 1 and the inequality $\ln(1 + x) \leq x$,

$$
\begin{aligned}
d_{\text{TV}}\left(\mathbf{A}, \mathcal{N}(0,1)^{m \times n}\right) &\leq O\left(\sqrt{D_2\left(\mathbf{A} \| \mathcal{N}(0,1)^{m \times n}\right)}\right) \\
&= O\left(\sqrt{\ln\left(\frac{\mathbb{E}_{\mathbf{A} \sim \mathcal{N}(0,1)^{m \times n}}\left[N(\mathbf{A})^2\right]}{\mathbb{E}_{\mathbf{A} \sim \mathcal{N}(0,1)^{m \times n}}\left[N(\mathbf{A})\right]^2}\right)}\right) \\
&\leq O\left(\sqrt{\ln\left(1 + O\left(\frac{\log n}{n}\right)\right)}\right) \\
&\leq O\left(\sqrt{\frac{\log n}{n}}\right),
\end{aligned}
$$

as desired.  $\square$

## B.4. Tightness

We show that the bounds in Theorem 2 and Theorem 3 are tight up to the log factors: The distance between the null and backdoored distributions is $\Omega(\sqrt{m/n})$, which is non-negligible. Moreover, the distinguisher that attains this advantage is efficient.

**Theorem 4.** *Assuming $\kappa^2 \leq 1/2$,*

$$
\Pr\left(\|\mathbf{A}\|_F^2 \leq mn - m/2\right) - \Pr\left(\|\mathcal{N}(0,1)^{m \times n}\|_F^2 \leq mn - m/2\right) = \Omega(\sqrt{m/n}).
$$

The random variable $\|\mathcal{N}(0,1)^{m \times n}\|_F^2$ is of type $\chi^2(mn)$, namely chi squared with $mn$ degrees of freedom.

Conditioned on $\mathbf{Ax} = \mathbf{y}$, $\|\mathbf{A}\|_F^2$ is of type $\chi^2(m(n-1)) + \|\mathbf{y}\|^2/n$. In particular, $\|\mathbf{A}\|_F^2$ is dominated by a random variable of type $\chi^2(mn - m) + \kappa^2 m$.

The reason is that an $n$-dimensional random normal vector $\mathbf{a}$ (representing a row of $\mathbf{A}$), when conditioned on a linear constraint $\mathbf{a}^\top \mathbf{x} = y$, projects to a standard normal in the $(n-1)$-dimensional subspace orthogonal to $\mathbf{x}$ and has fixed length $y/\|\mathbf{x}\| = y/\sqrt{n}$ in the direction of $\mathbf{x}$.

Thus $\|\mathbf{A}\|_F^2$ has mean at most $mn - (1 - \kappa^2)m$, while $\|\mathcal{N}(0,1)^{m \times n}\|_F^2$ has mean $mn$. The variance of both is (at most) $2mn$. Assuming they were sufficiently well-approximated by normals of the same mean and variance, their statistical distance would be on the order of $(1 - \kappa^2)m/\sqrt{2mn} = \Omega(\sqrt{m/n})$ as desired.

To complete the proof we argue that the error introduced by the normal approximation does not affect this estimate. The Berry-Esseen theorem gives an error term on the order of $1/\sqrt{mn}$. This completes the proof under the additional assumption that $m$ is at least some absolute constant.

To handle all values of $m$ including $m = 1$ we apply Cramér's first-order correction to the normal approximation of the chi squared CDF (Esseen, 1945; Pinelis, 2023):

$$
\Pr\left(\frac{\chi^2(k) - k}{\sqrt{2k}} \leq z\right) = \Pr(\mathcal{N}(0,1) \leq z) + \frac{\psi(z)}{\sqrt{k}} \pm O(1/k), \tag{5}
$$

where $\psi(z) = e^{-z^2/2} \cdot (1 - z^2)/3\sqrt{\pi}$.

*Proof.* The backdoored probability is at least

$$
\begin{aligned}
\Pr\left(\|\mathbf{A}\|_F^2 \leq mn - m/2\right) &\geq \Pr\left(\chi^2(mn - m) + \kappa^2 m \leq mn - m/2\right) &&\text{by domination} \\
&\geq \Pr\left(\frac{\chi^2(mn - m) - (mn - m)}{\sqrt{2(mn - m)}} \leq 0\right) &&\text{as } \kappa^2 \leq 1/2 \\
&= \frac{1}{2} + \frac{\psi(0)}{\sqrt{m(n-1)}} - O(1/mn). &&\text{by (5)}
\end{aligned}
$$

while the null probability is at most

$$\Pr\big(\|\mathcal{N}(0,1)^{m\times n}\|_F^2 \leq mn - m/2\big) = \Pr\left(\frac{\chi^2(mn) - mn}{\sqrt{2mn}} \leq -\frac{\sqrt{m/n}}{3\sqrt{2}}\right)$$

$$\leq \Pr\left(\mathcal{N}(0,1) \leq -\frac{\sqrt{m/n}}{3\sqrt{2}}\right) + \frac{\psi(0)}{\sqrt{mn}} + O(1/mn) \qquad \text{by (5)}$$

$$= \frac{1}{2} - \Omega(\sqrt{m/n}) + \frac{\psi(0)}{\sqrt{mn}} + O(1/mn)$$

as $\psi$ is maximized at zero. Thus the difference in probabilities is at least

$$\Omega(\sqrt{m/n}) - \psi(0)\left(\frac{1}{\sqrt{m(n-1)}} - \frac{1}{\sqrt{mn}}\right) - O(1/mn) = \Omega(\sqrt{m/n}) - O(1/mn + 1/m^{1/2}n^{3/2}).$$

The leading term $\Omega(\sqrt{m/n})$ dominates for all values of $m$. $\qquad\qquad\square$

## C. Constructing Backdoors for Neural Networks

### C.1. Defining Backdoors

Imagine that there is some learning procedure $\mathrm{ModelGen}()$ that generates some model $F$ (e.g., a neural network trained via stochastic gradient descent). To define the notion of an undetectable backdoor, we want the following properties to hold simultaneously:

- There is a way to generate a "backdoored" version of the model $F$, which gives anyone with $F$'s backdoor significant additional power over anyone without the backdoor.

- The "backdoored" model looks statistically close to an honest execution of $\mathrm{ModelGen}()$, in the sense that there is provably no distinguisher that works with high probability.

While the latter item is direct to formally define, the former requirement is vague. One possible way to specify such a requirement is via collision generation: it is hard to find collisions in an honest model $F$, but given a backdoor for $F$, one can easily compute collisions. By collisions, we mean distinct input vectors $\mathbf{x}_1, \mathbf{x}_2 \in \mathbb{Z}^n$ such that $F(\mathbf{x}_1) \approx F(\mathbf{x}_2)$. Our formal definition is below.

**Definition 6.** *A* statistically undetectable backdoor *with parameters $B$, $\delta_0$, and $\delta_1$ consists of p.p.t. algorithms* $(\mathrm{ModelGen}, \mathrm{BackdoorGen}, \mathrm{Activate})$ *with the following syntax:*

- $\mathrm{ModelGen}(1^n, 1^m)$. *This algorithm is randomized and outputs some model $F$.*

- $\mathrm{BackdoorGen}(1^n, 1^m)$. *This algorithm is randomized and outputs the tuple* $\left(\widehat{F}, \mathsf{bk}\right)$, *where $\widehat{F}$ is a backdoored model and $\mathsf{bk}$ is the corresponding backdoor key.*

- $\mathrm{Activate}(\mathsf{bk}, \mathbf{x} \in \mathbb{Z}^n)$. *This algorithm is deterministic and outputs a vector $\mathbf{x}' \in \mathbb{Z}^n$.*

*Moreover, we require the following three properties:*

1. *(**White Box**) **Statistical Undetectability**: For $F \leftarrow \mathrm{ModelGen}(1^n, 1^m)$ and $\left(\widehat{F}, \mathsf{bk}\right) \leftarrow \mathrm{BackdoorGen}(1^n, 1^m)$,*

$$d_{\mathrm{TV}}\left(F, \widehat{F}\right) = o(1).$$

*We emphasize that this total variation distance bound refers to the white-box descriptions of the models $F, \widehat{F}$.*

2. ***Backdoor Collision Generation**: For all $\mathbf{x} \in \mathbb{Z}^n$, it holds that*

$$\Pr_{\left(\widehat{F}, \mathsf{bk}\right) \leftarrow \mathrm{BackdoorGen}(1^n, 1^m)}\left(\mathbf{x}' \leftarrow \mathrm{Activate}\left(\mathsf{bk}, \mathbf{x}\right) : \begin{array}{l} \mathbf{x}' \in \mathbb{Z}^n, \|\mathbf{x}'\|_\infty \leq \|\mathbf{x}\|_\infty + 1, \\ \mathbf{x} \neq \mathbf{x}', \left\|\widehat{F}(\mathbf{x}') - \widehat{F}(\mathbf{x})\right\|_2 \leq \delta_0 \end{array}\right) = 1.$$

3. **Approximate Collision Resistance**: *For all p.p.t. algorithms $\mathcal{A}$,*

$$\Pr_{(\widehat{F},\mathsf{bk})\leftarrow\mathsf{BackdoorGen}(1^n,1^m)}\left((\mathbf{x}_1,\mathbf{x}_2)\leftarrow\mathcal{A}\left(\widehat{F}\right):\begin{array}{c}\mathbf{x}_1,\mathbf{x}_2\in[-B:B]^n,\\\mathbf{x}_1\neq\mathbf{x}_2,\left\|\widehat{F}(\mathbf{x}_2)-\widehat{F}(\mathbf{x}_1)\right\|_2\leq\delta_1\end{array}\right)=\mathsf{negl}(n),$$

*where the probability is also taken over the internal randomness of $\mathcal{A}$. We emphasize that $\mathcal{A}$ has white-box access to the model $\widehat{F}$ (e.g., its weights).*

*We define the* strength *of the backdoor be the quantity $\delta_1/\delta_0$, and we consider the backdoor meaningful only if $\delta_1/\delta_0 > 1$.*

This definition gives those with the backdoor additional power over others in two ways:

- Item 2 allows anyone with the backdoor to generate collisions for *all* inputs $\mathbf{x}$, while Item 3 stipulates hardness of finding even one collision.

- For $\delta_0 < \delta_1$ (as it is in our constructions), the backdoor generates collisions that are stronger than the impossibility bound for those without the backdoor. The larger the ratio $\delta_1/\delta_0$ is, the stronger this backdoor is, quantitatively. We call $\delta_1/\delta_0$ the *strength* of the backdoor for this reason.

While the condition in Item 2 that $\|\mathbf{x}'\|_\infty \leq \|\mathbf{x}\|_\infty + 1$ is somewhat arbitrary, the point is that the size of $\mathbf{x}'$ is similar to that of $\mathbf{x}$. One could formalize such a requirement in a few different ways, but we choose this one because it is what we achieve.

### C.2. Neural Network Preliminaries

Let $\mathbf{A} \in \mathbb{R}^{m_2\times m_1}$. We let $\sigma_{\max}(\mathbf{A})$ denote the maximum singular value of $\mathbf{A}$, and we let $\sigma_{\min}(\mathbf{A})$ denote the minimum singular value of $\mathbf{A}$. More explicitly,

$$\sigma_{\max}(\mathbf{A}) = \sup_{\mathbf{x}\in\mathbb{R}^{m_1}\setminus\{\mathbf{0}\}}\frac{\|\mathbf{A}\mathbf{x}\|_2}{\|\mathbf{x}\|_2},$$

$$\sigma_{\min}(\mathbf{A}) = \inf_{\mathbf{x}\in\mathbb{R}^{m_1}\setminus\{\mathbf{0}\}}\frac{\|\mathbf{A}\mathbf{x}\|_2}{\|\mathbf{x}\|_2}.$$

Note that if $m_1 > m_2$, then $\sigma_{\min}(\mathbf{A}) = 0$, as $\mathbf{A}$ has a nontrivial kernel. Whenever $\sigma_{\min}(\mathbf{A}) > 0$, we can let $\mathrm{cond}(\mathbf{A})$ denote the condition number of $\mathbf{A}$, defined as

$$\mathrm{cond}(\mathbf{A}) = \frac{\sigma_{\max}(\mathbf{A})}{\sigma_{\min}(\mathbf{A})} \geq 1. \tag{6}$$

**Definition 7** (Bi-Lipschitz Functions). *For $m_1, m_2 \in \mathbb{N}$ and $0 \leq \alpha \leq \beta$, we say a function $f : \mathbb{R}^{m_1} \to \mathbb{R}^{m_2}$ is $(\alpha, \beta)$-bilipschitz if for all $\mathbf{x}, \mathbf{y} \in \mathbb{R}^{m_1}$,*

$$\alpha\|\mathbf{x} - \mathbf{y}\|_2 \leq \|f(\mathbf{x}) - f(\mathbf{y})\|_2 \leq \beta\|\mathbf{x} - \mathbf{y}\|_2.$$

*Moreover, for $\xi \geq 1$, we say $f$ has* distortion *at most $\xi$ if there exist $\beta \geq \alpha \geq 0$ such that $f$ is $(\alpha, \beta)$-bilipschitz and $\xi = \beta/\alpha$.*

**Fact 2.** *Suppose $f_1 : \mathbb{R}^{m_1} \to \mathbb{R}^{m_2}$ and $f_2 : \mathbb{R}^{m_2} \to \mathbb{R}^{m_3}$ are $(\alpha_1, \beta_1)$-bilipschitz and $(\alpha_2, \beta_2)$-bilipschitz, respectively. Then $f_2 \circ f_1 : \mathbb{R}^{m_1} \to \mathbb{R}^{m_3}$ is $(\alpha_1\alpha_2, \beta_1\beta_2)$-bilipschitz.*

**Fact 3.** *For a matrix $\mathbf{A} \in \mathbb{R}^{m_2\times m_1}$, the linear map given by $\mathbf{A}$, mapping $\mathbb{R}^{m_1}$ to $\mathbb{R}^{m_2}$, is $(\sigma_{\min}(\mathbf{A}), \sigma_{\max}(\mathbf{A}))$-bilipschitz.*

**Definition 8.** *For $\alpha \in (0, 1)$, the* leaky rectified linear unit (leaky ReLU) *with parameter $\alpha$ is the function $\mathrm{LeakyReLU}_\alpha : \mathbb{R} \to \mathbb{R}$ defined by*

$$\mathrm{LeakyReLU}_\alpha(x) = \begin{cases} x & x > 0, \\ \alpha x & x \leq 0. \end{cases}$$

*To slightly abuse notation, it naturally generalizes to a function $\mathrm{LeakyReLU}_\alpha : \mathbb{R}^m \to \mathbb{R}^m$ where (the scalar version of) $\mathrm{LeakyReLU}_\alpha$ is applied coordinate-wise.*

**Fact 4.** *For all $\alpha \in (0,1)$ and for all $m \in \mathbb{N}$, $\text{LeakyReLU}_\alpha : \mathbb{R}^m \to \mathbb{R}^m$ is $(\alpha, 1)$-bilipschitz.*

For depth $d \in \mathbb{N}$, a feedforward neural network is defined in terms of weight matrices $\mathbf{A}^{(0)}, \cdots, \mathbf{A}^{(d-1)}$, bias vectors $\mathbf{b}^{(0)}, \cdots, \mathbf{b}^{(d-1)}$, and an activation function $\sigma : \mathbb{R} \to \mathbb{R}$. The mapping takes in a vector $\mathbf{x} = \mathbf{x}^{(0)}$, iteratively evaluates

$$\mathbf{x}^{(i+1)} := \sigma\left(\mathbf{A}^{(i)}\mathbf{x}^{(i)} + \mathbf{b}^{(i)}\right),$$

and outputs $\mathbf{x}^{(d)}$, where $\sigma$ is applied pointwise. The matrices $\mathbf{A}^{(i)}$ can be rectangular (instead of square) with the constraint that the input vector $\mathbf{x}$, bias vectors $\mathbf{b}^{(i)}$, and weight matrices $\mathbf{A}^{(i)}$ all have dimensions that syntactically align.

**Lemma 4.** *For $\alpha \in (0,1)$, a feedforward neural network of depth $d$ with weight matrices $\mathbf{A}^{(0)}, \cdots, \mathbf{A}^{(d-1)}$, bias vectors $\mathbf{b}^{(0)}, \cdots, \mathbf{b}^{(d-1)}$, and activation function $\text{LeakyReLU}_\alpha$ is $(\alpha', \beta')$-bilipschitz, where*

$$\alpha' = \alpha^d \prod_{i=0}^{d-1} \sigma_{\min}\left(\mathbf{A}^{(i)}\right),$$

$$\beta' = \prod_{i=0}^{d-1} \sigma_{\max}\left(\mathbf{A}^{(i)}\right).$$

*Moreover, if one skips the first layer matrix $\mathbf{A}^{(0)}$ and directly applies the activation function to the input vector $\mathbf{x}$ (and then $\mathbf{A}^{(1)}$ and so on), the resulting function is $(\alpha', \beta')$-bilipschitz, where*

$$\alpha' = \alpha^d \prod_{i=1}^{d-1} \sigma_{\min}\left(\mathbf{A}^{(i)}\right),$$

$$\beta' = \prod_{i=1}^{d-1} \sigma_{\max}\left(\mathbf{A}^{(i)}\right).$$

*Proof.* This follows by directly combining Fact 3, Fact 4, and Fact 2 and layer-by-layer induction, as addition by any bias vector $\mathbf{b}^{(i)}$ is a $(1, 1)$-bilipschitz operation. $\qquad\square$

### C.3. Construction

The most general template for our backdoor construction will be as follows. Let $\mathbf{A} \sim \mathcal{N}(0,1)^{m \times n}$, and let $\mathcal{T}$ be any (randomized) training operator that takes in $\mathbf{A} \in \mathbb{R}^{m \times n}$ and outputs an $(\alpha, \beta)$-bilipschitz function $g \leftarrow \mathcal{T}(\mathbf{A})$. We will construct backdoors for the model class given by

$$F(\mathbf{x}) := g(\mathbf{A}\mathbf{x}).$$

The backdoor construction is direct: generate $\left(\widehat{\mathbf{A}}, \mathbf{z}\right) \leftarrow \text{BackdoorMatrix}(1^n, 1^m)$, and to activate any $\mathbf{x}$, output $\mathbf{x}' = \mathbf{x} + \mathbf{z}$. By linearity, $\mathbf{A}\mathbf{x}' = \mathbf{A}(\mathbf{x} + \mathbf{z}) = \mathbf{A}\mathbf{x} + \mathbf{A}\mathbf{z} \approx \mathbf{A}\mathbf{x}$, and by lipschitzness of $g$,

$$F(\mathbf{x}') = g(\mathbf{A}\mathbf{x}') \approx g(\mathbf{A}\mathbf{x}) = F(\mathbf{x}).$$

Conversely, if a p.p.t. algorithm computes $\mathbf{x}_1 \neq \mathbf{x}_2 \in [-B : B]^n$ such that $F(\mathbf{x}_1) \approx F(\mathbf{x}_2)$, then by bilipschitzness of $g$, it follows that $\mathbf{A}\mathbf{x}_1 \approx \mathbf{A}\mathbf{x}_2$, and therefore $\mathbf{A}(\mathbf{x}_1 - \mathbf{x}_2) \approx \mathbf{0}$, violating Assumption 2. We give the formal statement in Theorem 5.

---

Generic Backdoor Construction

- ModelGen($1^n, 1^m$): Sample $\mathbf{A} \sim \mathcal{N}(0,1)^{m \times n}$, sample $g \leftarrow \mathcal{T}(\mathbf{A})$, define the model

$$F(\mathbf{x}) = g(\mathbf{A}\mathbf{x}),$$

  and output the description of the model $F$.

- BackdoorGen($1^n, 1^m$): Sample $\left(\widehat{\mathbf{A}}, \mathbf{z}\right) \leftarrow \text{BackdoorMatrix}(1^n, 1^m)$, sample $\widehat{g} \leftarrow \mathcal{T}\left(\widehat{\mathbf{A}}\right)$, define the model

$$\widehat{F}(\mathbf{x}) = \widehat{g}\left(\widehat{\mathbf{A}}\mathbf{x}\right),$$

  and output $\left(\widehat{F}, \text{bk} = \mathbf{z}\right)$.

- Activate(bk, $\mathbf{x}$): Parsing $\mathbf{z} = \text{bk}$, output $\mathbf{x} + \mathbf{z}$.

---

*Figure 5.* The generic construction of backdoors for linear models with bilipschitz postprocessing, as used in Theorems 5 and 6.

**Theorem 5.** *For all $m = n^{\Omega(1)}$ and $m = o(n)$, consider* ModelGen($1^n, 1^m$) *to output models of the form*

$$F(\mathbf{x}) = g(\mathbf{A}\mathbf{x}),$$

*where $\mathbf{A} \sim \mathcal{N}(0,1)^{m \times n}$ and $g \leftarrow \mathcal{T}(\mathbf{A})$, where $\mathcal{T}$ is a p.p.t. training operator supported only on $(\alpha, \beta)$-bilipschitz functions. Then, for all $B \leq \text{poly}(n)$, under Assumption 2, Figure 5 gives a statistically undetectable backdoor for* ModelGen *with parameters $B$ and*

$$\delta_0 = O\left(\frac{\beta\sqrt{m}}{2^{n/m}}\right), \quad \delta_1 = \Omega\left(\frac{\alpha}{m^{\varepsilon}\sqrt{n}}\right),$$

*for all $\varepsilon > 0$. In particular, the strength of the backdoor is*

$$\frac{\delta_1}{\delta_0} = \Omega\left(\frac{\alpha \cdot 2^{n/m}}{\beta\sqrt{n} \cdot m^{1/2+\varepsilon}}\right).$$

We also state a version where $m = 1$.

**Theorem 6.** *For $m = 1$, consider* ModelGen($1^n$) *to output models of the form*

$$F(\mathbf{x}) = g\left(\mathbf{a}^{\top}\mathbf{x}\right),$$

*where $\mathbf{a} \sim \mathcal{N}(0,1)^n$ and $g \leftarrow \mathcal{T}(\mathbf{A})$, where $\mathcal{T}$ is a p.p.t. training operator supported only on $(\alpha, \beta)$-bilipschitz functions. Then, there exists a universal constant $C > 0$ such that for all $B \leq \text{poly}(n)$ and $\varepsilon > 0$, under Assumption 1, Figure 5 gives a statistically undetectable backdoor for* ModelGen *with parameters $B$ and*

$$\delta_0 = O\left(\frac{\beta \cdot n^C}{2^n}\right), \quad \delta_1 = \frac{\alpha}{2^{O\left(\log^{3+\varepsilon}(n)\right)}}.$$

*In particular, the strength of the backdoor is*

$$\frac{\delta_1}{\delta_0} = \frac{\alpha \cdot 2^n}{\beta \cdot 2^{O\left(\log^{3+\varepsilon} n\right)}},$$

*for all $\varepsilon > 0$.*

*Proof of Theorem 5.* The construction is given in Figure 5. We prove each of the properties in turn.

To see statistical indistinguishability, note that

$$d_{\text{TV}}\left(\widehat{\mathbf{A}}, \mathcal{N}(0,1)^{m \times n}\right) = o(1)$$

by Theorem 2. Since $\mathrm{ModelGen}$ and $\mathrm{BackdoorGen}$ are random processes that differ only in how the matrices are sampled, the data processing inequality implies

$$d_{\mathrm{TV}}\left(F, \widehat{F}\right) = o(1),$$

as desired.

To see backdoor collision generation, recall that

$$\left\|\widehat{\mathbf{A}}\mathbf{z}\right\|_{\infty} \leq O\left(\frac{n}{2^{n/m}}\right)$$

by Theorem 2. Clearly $\mathbf{x}' = \mathrm{Activate}(\mathrm{bk}, \mathbf{x}) = \mathbf{x} + \mathbf{z} \in \mathbb{Z}^n$, $\mathbf{x}' \neq \mathbf{x}$, and $\|\mathbf{x}'\|_{\infty} \leq \|\mathbf{x}\|_{\infty} + 1$, so it suffices to show that

$$\left\|\widehat{F}(\mathbf{x}') - \widehat{F}(\mathbf{x})\right\|_{2} \leq \delta_0.$$

We have

$$\left\|\widehat{F}(\mathbf{x}') - \widehat{F}(\mathbf{x})\right\|_{2} = \left\|\widehat{g}\left(\widehat{\mathbf{A}}\mathbf{x}'\right) - \widehat{g}\left(\widehat{\mathbf{A}}\mathbf{x}\right)\right\|_{2} = \left\|\widehat{g}\left(\widehat{\mathbf{A}}\mathbf{x} + \widehat{\mathbf{A}}\mathbf{z}\right) - \widehat{g}\left(\widehat{\mathbf{A}}\mathbf{x}\right)\right\|_{2}$$

$$\leq \beta \cdot \left\|\widehat{\mathbf{A}}\mathbf{z}\right\|_{2}$$

$$\leq \beta\sqrt{m} \cdot \left\|\widehat{\mathbf{A}}\mathbf{z}\right\|_{\infty}$$

$$\leq O\left(\frac{\beta\sqrt{m}}{2^{n/m}}\right).$$

Therefore, we can set $\delta_0 = O(\beta\sqrt{m} \cdot 2^{-n/m})$.

Finally, to see approximate collision resistance, suppose for contradiction that there exists a p.p.t. algorithm $\mathcal{A}$ and a constant $C > 0$ such that

$$\Pr_{\left(\widehat{F},\mathrm{bk}\right)\leftarrow\mathrm{BackdoorGen}(1^n, 1^m)}\left((\mathbf{x}_1, \mathbf{x}_2) \leftarrow \mathcal{A}\left(\widehat{F}\right) : \begin{array}{c} \mathbf{x}_1, \mathbf{x}_2 \in [-B : B]^n, \\ \mathbf{x}_1 \neq \mathbf{x}_2, \left\|\widehat{F}(\mathbf{x}_2) - \widehat{F}(\mathbf{x}_1)\right\|_{2} \leq \delta_1 \end{array}\right) \geq \frac{1}{n^C},$$

for infinitely many values of $n$. Consider an algorithm $\mathcal{A}'$ (using $\mathcal{A}$) defined as follows: On input a matrix $\mathbf{A} \in \mathbb{R}^{m \times n}$, sample $g \leftarrow \mathcal{T}(\mathbf{A})$, define $F(\mathbf{x}) = g(\mathbf{A}\mathbf{x})$, and receive $(\mathbf{x}_1, \mathbf{x}_2) \leftarrow \mathcal{A}(F)$. The algorithm $\mathcal{A}'$ then outputs $\mathbf{x}_1 - \mathbf{x}_2 \in [-2B : 2B]^n \setminus \{0^n\}$. The claim is that the p.p.t. algorithm $\mathcal{A}'$ violates Assumption 2. To see this, note that

$$\left\|\widehat{F}\left(\mathbf{x}_2\right) - \widehat{F}\left(\mathbf{x}_1\right)\right\|_{2} \leq \delta_1 \iff \left\|\widehat{g}\left(\widehat{\mathbf{A}}\mathbf{x}_2\right) - \widehat{g}\left(\widehat{\mathbf{A}}\mathbf{x}_1\right)\right\|_{2} \leq \delta_1$$

$$\implies \left\|\widehat{\mathbf{A}}\mathbf{x}_2 - \widehat{\mathbf{A}}\mathbf{x}_1\right\|_{2} \leq \frac{\delta_1}{\alpha}$$

$$\implies \left\|\widehat{\mathbf{A}}\mathbf{x}_2 - \widehat{\mathbf{A}}\mathbf{x}_1\right\|_{\infty} \leq \frac{\delta_1}{\alpha}$$

Therefore, we have the following:

$$\Pr_{\left(\widehat{\mathbf{A}},\mathbf{z}\right)\leftarrow\mathrm{BackdoorMatrix}(1^n, 1^m)}\left(\mathbf{x} \leftarrow \mathcal{A}'\left(\widehat{\mathbf{A}}\right) : \begin{array}{c} \mathbf{x} \in [-2B : 2B]^n \setminus \{\mathbf{0}\}, \\ \left\|\widehat{\mathbf{A}}\mathbf{x}\right\|_{\infty} \leq \delta_1/\alpha \end{array}\right) \geq \frac{1}{n^C},$$

for infinitely many values of $n$. Let $E = E(\mathbf{A})$ denote the above event (as a function of matrix $\mathbf{A}$), so that

$$\Pr_{\left(\widehat{\mathbf{A}},\cdot\right)\leftarrow\mathrm{BackdoorMatrix}(1^n, 1^m)}\left(E\left(\widehat{\mathbf{A}}\right)\right) \geq \frac{1}{n^C}$$

infinitely often. By Lemma 3 and Rényi closeness of $\widehat{\mathbf{A}}$ and $\mathbf{A} \sim \mathcal{N}(0,1)^{m \times n}$ (as guaranteed by Theorem 2) we have

$$\Pr_{\mathbf{A}\sim\mathcal{N}(0,1)^{m \times n}}\left(E\left(\mathbf{A}\right)\right) \geq \frac{\Pr_{\left(\widehat{\mathbf{A}},\cdot\right)\leftarrow\mathrm{BackdoorMatrix}(1^n, 1^m)}\left(E\left(\widehat{\mathbf{A}}\right)\right)^2}{e^{D_2\left(\widehat{\mathbf{A}}||\mathbf{A}\right)}}$$

$$\geq \frac{1/n^{2C}}{e^{o(1)}} = \Omega\left(\frac{1}{n^{2C}}\right)$$

infinitely often. That is,

$$\Pr_{\mathbf{A} \sim \mathcal{N}(0,1)^{m \times n}} \left( \mathbf{x} \leftarrow \mathcal{A}'(\mathbf{A}) : \begin{array}{l} \mathbf{x} \in [-2B : 2B]^n \setminus \{\mathbf{0}\}, \\ \|\mathbf{A}\mathbf{x}\|_\infty \leq \delta_1/\alpha \end{array} \right) = \Pr_{\mathbf{A} \sim \mathcal{N}(0,1)^{m \times n}} (E(\mathbf{A})) = \Omega\left(\frac{1}{n^{2C}}\right),$$

for infinitely many values of $n$. By the parameters of Assumption 2, we can set $\delta_1 = \alpha/(m^\varepsilon \sqrt{n})$ for any $\varepsilon > 0$ to arrive at the contradiction. $\qquad\square$

*Proof of Theorem 6.* The proof is exactly that of Theorem 5, with the difference being that we apply Theorem 3 instead of Theorem 2 and Assumption 1 instead of Assumption 2. This changes the bound of $\delta_0$ to $\delta_0 = O(\beta \cdot 2^{-n} \cdot n^C)$, and similarly, $\delta_1 = \alpha/2^{\Theta(\log^{3+\varepsilon}(n))}$. $\qquad\square$

## C.4. Backdoors in Deep Neural Networks

Here, we combine Appendices C.2 and C.3 to show how to insert backdoors in certain architectures of deep feedforward neural networks.

- The first linear layer needs to be a random compressing Gaussian matrix $\mathbf{A} \sim \mathcal{N}(0,1)^{m \times n}$ (where $n \gg m$). This is a common paradigm in random feature learning (Rahimi & Recht, 2007).

- The activation function needs to be bilipschitz.

- The linear maps in the second layer and onward need to be well-conditioned, in the sense that

$$\mathrm{cond}(\mathbf{A}) = \frac{\sigma_{\max}(\mathbf{A})}{\sigma_{\min}(\mathbf{A})} \approx 1,$$

  with flexibility on the distance from 1. Note that such linear maps can either be dimension-preserving or expanding.

More precisely, let $\mathsf{NN}_{n,d,m,\alpha,\gamma}$ denote the following class of depth-$d$ feedforward neural networks:

- The first linear layer $\mathbf{A}^{(0)} \sim \mathcal{N}(0,1)^{m \times n}$ is a random $m \times n$ Gaussian matrix that is unchanged throughout training, where $m$ and $n$ are parameters.

- The linear maps $\mathbf{A}^{(1)}, \mathbf{A}^{(2)}, \cdots, \mathbf{A}^{(d-1)}$ are arbitrary but well-conditioned, in the sense that for all $i \in \{1, 2, \cdots, d-1\}$,

$$\mathrm{cond}\left(\mathbf{A}^{(i)}\right) \leq \gamma$$

  where $\gamma \geq 1$ is a parameter. In particular, $\mathbf{A}^{(1)}, \cdots, \mathbf{A}^{(d-1)}$ can all be updated throughout training, as long as they end up not being too ill-conditioned.

- All activation functions $\sigma : \mathbb{R} \to \mathbb{R}$ are $\mathrm{LeakyReLU}_\alpha$, where $\alpha \in (0,1)$ is a parameter.

**Theorem 7.** *For $m = n^{\Omega(1)}$ and $m = o(n)$, and for any parameters $d \in \mathbb{N}$, $\alpha \in (0,1)$, $\gamma \geq 1$ let $\mathrm{ModelGen}(1^n, 1^m)$ output neural networks that are in $\mathsf{NN}_{n,d,m,\alpha,\gamma}$. For all $B \leq \mathrm{poly}(n)$, under Assumption 2, there exists a statistically undetectable backdoor for $\mathrm{ModelGen}$ with strength*

$$\frac{\delta_1}{\delta_0} = \Omega\left(\frac{\alpha^d \cdot 2^{n/m}}{\sqrt{n} \cdot m^{1/2+\varepsilon} \cdot \gamma^{d-1}}\right),$$

*for all $\varepsilon > 0$.*

*Proof of Theorem 7.* We directly apply Theorem 5, where $\mathcal{T}$ neural networks as described in NN except skipping the first layer $\mathbf{A}^{(0)}$. By Lemma 4, we know that $\mathcal{T}$ is supported on $(\alpha', \beta')$-bilipschitz functions, where

$$\alpha' = \alpha^d \prod_{i=1}^{d-1} \sigma_{\min}\left(\mathbf{A}^{(i)}\right),$$

$$\beta' = \prod_{i=1}^{d-1} \sigma_{\max}\left(\mathbf{A}^{(i)}\right).$$

Plugging this into Theorem 5, the strength of the backdoor is

$$\frac{\delta_1}{\delta_0} = \Omega\left(\frac{\alpha' \cdot 2^{n/m}}{\beta'\sqrt{n} \cdot m^{1/2+\varepsilon}}\right) = \Omega\left(\frac{\alpha^d \cdot 2^{n/m} \prod_{i=1}^{d-1} \sigma_{\min}\left(\mathbf{A}^{(i)}\right)}{\sqrt{n} \cdot m^{1/2+\varepsilon} \prod_{i=1}^{d-1} \sigma_{\max}\left(\mathbf{A}^{(i)}\right)}\right)$$

$$= \Omega\left(\frac{\alpha^d \cdot 2^{n/m}}{\sqrt{n} \cdot m^{1/2+\varepsilon} \prod_{i=1}^{d-1} \mathrm{cond}\left(\mathbf{A}^{(i)}\right)}\right)$$

$$= \Omega\left(\frac{\alpha^d \cdot 2^{n/m}}{\sqrt{n} \cdot m^{1/2+\varepsilon} \cdot \gamma^{d-1}}\right),$$

for all $\varepsilon > 0$, as desired. $\qquad\square$

We now instantiate Theorem 7 with slightly more concrete parameter choices. The reason for setting $\alpha \geq 1/100$ for the LeakyReLU is that $\alpha = 1/100$ is a commonly used default value, e.g., in PyTorch (Paszke et al., 2019).

**Corollary 3.** *For $m = n^{1/2}$, $d = n^{1/4}$, any $\alpha \in [1/100, 1)$, and any $\gamma \in \left[1, 2^{n^{1/5}}\right]$, under Assumption 2, for all $B \leq \mathrm{poly}(n)$, there exists a statistically undetectable backdoor for $\mathsf{NN}_{n,d,m,\alpha,\gamma}$ with strength*

$$\frac{\delta_1}{\delta_0} = 2^{\Omega(m)}.$$

*Proof.* We directly plug these parameters into Theorem 7 (and $\varepsilon = 1/2$) to get strength

$$\frac{\delta_1}{\delta_0} = \Omega\left(\frac{\alpha^d \cdot 2^{n/m}}{\sqrt{n} \cdot m^{1/2+\varepsilon} \cdot \gamma^{d-1}}\right)$$

$$= \Omega\left(\frac{2^{\sqrt{n}}}{100^{n^{1/4}} \cdot n^{1/2} \cdot (2^{n^{1/5}})^{n^{1/4}-1}}\right)$$

$$= \Omega\left(\frac{2^{\sqrt{n}}}{2^{O(n^{9/20})}}\right)$$

$$= 2^{\Omega(\sqrt{n})}.$$

$\qquad\square$

# Use of Large Language Models

We used large language models (specifically, Claude Code) to help generate code for our implementation as done in Section 3.1.

