# OpenReview forum: "Statistically Undetectable Backdoors in Deep Neural Networks"
_ICML.cc/2026/Conference — ICML 2026 regular_

### Official Review · Reviewer_wm2G · 2026-03-06

**Soundness:** 3
**Presentation:** 1
**Significance:** 3
**Originality:** 2
**Overall Recommendation:** 5
**Confidence:** 3

**Summary:**

This paper studies invariance-based backdoors, which aim to get similar outputs for two inputs $x$ and $x+z$, where $z$ is the backdoor. Utilizing the Matrix Backdoor Construction (MBC) algorithm and the cryptographical security behind MBC, the paper shows that other than the creator with the backdoor, no efficient (polynomial-time) adversary can generate two examples $x$ and $x'$ that are close, which means that no efficient adversary can generate $x$ and $x'$ with similar outputs under the bi-Lipschitz condition. By upper-bounding the TV-distance between the standard multi-dimensional Gaussian and the distribution of the matrix $A$ generated in MBC, the paper shows that the injected backdoor is statistically undetectable. The paper also conducts experiments accordingly.

**Compliance With Llm Reviewing Policy:**

Affirmed.

**Final Justification:**

The rebuttal addressed my concerns, and I choose to increase my score from 4 to 5.

**Key Questions For Authors:**

- Please refer to the weaknesses.
- The theoretical results are interesting and novel. However, I am not very sure about the originality of the theoretical results, and I need the authors to clarify it. The main challenge of the paper lies in the analysis of the total variation distance between the distribution of the planted matrix and a truly Gaussian one. My question is: as a classical cryptographic algorithm, is there any work that analyzes the distributions of $A$ generated by the Matrix Backdoor Construction algorithm or the distance (not necessarily TV-distance) between $A$ and $\mathcal{N}(0,I)$?

**Limitations:**

NA.

**Strengths And Weaknesses:**

### Strengths
- The theoretical results for statistically undetectable invariance-based backdoors are very novel.


### Weaknesses
- Although this is a theoretical paper, the Fashion-MNIST dataset is too small. Experiments on larger datasets like CIFAR are recommended. However, it is OK if the authors do not think it is necessary to do such experiments. This point does not have an influence on my final evaluation.
- Although the paper justified Constraint 1 in section 1.1, it is unclear whether a frozen random compressing will affect the performance of a neural network. I believe this should be discussed in detail.
- It is not certain whether adding regularizations to make sure the bi-Lipschitz condition holds will affect the performance of a neural network.
- It is clear that invariance-based backdoors can be used to establish ownership. However, there seems to lack of examples about how to use invariance-based backdoors to attack a model in the reference stage.
- The readability of the paper is not so good. There are many undefined terms (before used) and confusing expressions in the main content, which is not friendly to readers. For example:
  - In Theorem 7, it is unclear what the paper means by "The total variation distance between the descriptions of $M_\mathcal{A}$ and $M_\mathcal{B}$". The readers have to understand this point after going through the whole appendix, and I am not sure whether I wrongly understood it since I did not check the proof step by step. It seems that this term means: if we do not inject a backdoor, we use $A \sim \mathcal{N}(0, I)$; if we inject a backdoor, we use $A'$ sampled from MBC. The term means the TV-distance between the distributions of $A$ and $A'$.
  - In Theorem 7, the term $\beta_ \text{upper}$ is undefined.
- If "The total variation distance between the descriptions of $M_\mathcal{A}$ and $M_\mathcal{B}$" is the TV-distance between the distributions of $A$ and $A'$ as I mentioned above, then the statement in lines 157-160 (left column) is not obvious, and the authors might need to prove it.

---

> ### Author Rebuttal · Authors · 2026-03-31
>
> Thank you for your positive review. We’re glad that you found our theoretical results interesting and very novel. We respond to your review below, and we hope that our response adequately addresses your questions and concerns. If not, please let us know so that we can appropriately respond.
>
> **Performance concerns with Constraints 1 and 2**:
>
> Whether Constraints 1 and 2 affect model performance is an empirical question that is largely beyond the scope of the submission. Nonetheless, we provide justification in Section 1.1 that these constraints are reasonable and have been previously studied. There is a rich body of literature stemming from the JL lemma and random feature learning (Rahimi & Recht, 2007) that all make compelling arguments for the paradigm of Constraint 1 (random compressing Gaussian matrix). For Constraint 2, we want to emphasize that some works even show quality *improvements* when enforcing Lipschitzness (e.g., (Yoshida & Miyato, 2017; Miyato et al., 2018)).
>
> **Example of backdoor use**:
>
> Below, we present a scenario in which the backdoor notion we construct is significant and potentially worrisome (instead of for establishing ownership). Imagine that one trains a DNN as an image embedding model used within a facial recognition system. Let $M$ denote the model given as a DNN. To recognize a new image $\mathbf{x}$, one would evaluate $M(\mathbf{x})$ and compute the distances between $M(\mathbf{x})$ and $M(\mathbf{x}’)$ for target images $\mathbf{x}’$ and apply a threshold. If there is an existing target image $\mathbf{x}’$ such that $\| M(\mathbf{x}) - M(\mathbf{x}’) \|$ is sufficiently small, the input is classified as a “match” and the system recognizes the image.
>
> Our backdoor notion in this setting would mean the following. With the backdoor, one can take any image and generate a fake, different image that is a “fake match” with the original image according to this recognition system. Conversely, without the backdoor, no one else has this ability to “fool” the recognition system, even for one image.
>
> **Readability concerns**:
>
> Regarding "the total variation distance between the descriptions of $M_{\mathcal{A}}$ and $M_{\mathcal{B}}$", we mean that we are not merely referring to these models as functions, but also as lists of explicit DNN parameters (e.g., the collection of weights and biases). We mean more than just TV-distance between $A$ and $A’$; we mean closeness of all parameters in the DNN, considered as a joint distribution. In particular, this implies that the distributions of $M_{\mathcal{A}}$ and $M_{\mathcal{B}}$ are close in TV-distance as functions as well.
>
> The parameter $\beta_{\mathrm{upper}}$ (as used in Theorem 7) is specified earlier in Constraint 2 as a bound on the bi-Lipschitzness of the remaining layers of the network.
>
> Thank you for letting us know about your confusion on both of these points; we are grateful. In the revision, we will clarify the writing to remove any ambiguity. Please let us know if you still have any remaining questions about these points.
>
> **MBC analysis originality**:
>
> We do not consider our MBC algorithm a “classical cryptographic algorithm”: the algorithm is novel (as far as we are aware). In particular, there have been no prior attempts to analyze the distance, TV or otherwise.

---

> > ### Author Rebuttal · Reviewer_wm2G · 2026-04-02
> >
> > I would like to thank the authors for the detailed reply. Some of my concerns are addressed, and my remaining concerns are as follows.
> >
> > **About the example of backdoor use.** I am sorry I did not make it clear. My concern is that the proposed method in this paper is limited in the data poisoning backdoor attack situation, where the attacker can only manipulate the training data but not the network structure. Unless the user just gets a trained network from the agents, I do not see the possibility of injecting a backdoor. However, "attacking the network I trained" is somehow a strange setting. In this way, Constraint 1 is a little strong when it comes to backdoor attacks.
> >
> > **About the MBC analysis originality.** You said that "We do not consider our MBC algorithm a classical cryptographic algorithm: the algorithm is novel (as far as we are aware)". Do you mean that the MBC algorithm is proposed and first analyzed by this paper?
> >
> > **------- update after the authors' reply -------**
> >
> > Thank you for clarifying your contribution. My questions are resoved and I am will increase my score to 5.

---

> > > ### Author Response · Authors · 2026-04-08
> > >
> > > Thank you for your acknowledgement. We are grateful for your follow-up clarifications.
> > >
> > > **Example of backdoor use**:
> > >
> > > We agree that our backdoor setup does not naturally fit into a data poisoning paradigm. That being said, we would like to clarify that our result can be interpreted as saying that it is sufficient to backdoor only the *randomness* used at the start of training, without needing to tamper with training data or gradient descent. Randomness generation is a complex and subtle task, and as such, it is often delegated to particular libraries or secure hardware implementations. For example, we would imagine most practitioners trust that the function “numpy.random.normal()” outputs a clean Gaussian sample without any adversarial auxiliary information. If a malicious party were to implement this function in a way that gives them access to a backdoor (as in our paper), then they could perhaps have broad access to backdoors in DNNs. While such threats may seem hypothetical, in cryptography standardization there have been famous and devastating instances of malicious randomness generation. For example, the Dual_EC_DRBG algorithm is a pseudorandom number generator that was standardized by the National Institute of Standards and Technology (NIST), but it was later discovered that the National Security Agency (NSA) planted a secret backdoor in it.
> > >
> > > **MBC analysis originality**:
> > >
> > > Yes, we mean that the MBC algorithm is proposed and first analyzed by this paper (as far as we are aware).

---

### Official Review · Reviewer_2crB · 2026-03-08

**Soundness:** 3
**Presentation:** 3
**Significance:** 3
**Originality:** 3
**Overall Recommendation:** 4
**Confidence:** 3

**Summary:**

This paper shows that an adversarial model trainer can inject backdoors in feedforward deep neural networks such that the backdoored model is statistically indistinguishable from an honestly trained model, even in the white-box setting where all weights are visible. The backdoor enables the trainer to produce invariance-based adversarial examples for any input, while no efficient adversary can produce any such colliding pair without the backdoor. The construction works by rejection-sampling the first-layer Gaussian matrix such that a secret vector $z$ yields very small output under the matrix, leveraging the computational hardness of the symmetric binary perceptron or number balancing problem. The key technical contribution is bounding the second-moment concentration of the number of $\pm1$ solutions to show that the planted matrix remains statistically close to a truly random Gaussian. An empirical experiment on Fashion-MNIST embeddings and preliminary experiments are conducted.

**Compliance With Llm Reviewing Policy:**

Affirmed.

**Final Justification:**

My concerns are addressed and I will maintain my score

**Key Questions For Authors:**

Please refer to the weakness

**Limitations:**

yes

**Strengths And Weaknesses:**

**Strength**

- The paper introduces the backdoor strength as a measure of the power asymmetry between trainer and adversary, and proves exponential backdoor strength under standard cryptographic assumptions, which is a novel and interesting.
- Unlike the literature that only achieves computational indistinguishability, this construction achieves closeness in TV distance, meaning no algorithm can distinguish the backdoored model from an honest one with more than a small advantage. This is a stronger guarantee.
- The overall writing of this paper is clear and easy to follow.

**Weakness**

- Constraints 1 and 2 seem quite restrictive for modern architectures. The authors acknowledge this but most practical DNNs do not use frozen random first layers, and requiring all subsequent layers to be expanding or square precludes standard classification heads.
- The experiments are presented as proof-of-concept, but they remain limited. The Fashion-MNIST embedding model is small and there is no comparison against existing backdoor detection methods. The computational hardness experiments only go up to $n=100$, $m=30$, and test only three simple algorithms. A more thorough empirical study with larger models, more datasets and baselines would significantly strengthen the paper.
- The total variation bound $\epsilon = O(\sqrt{m/n}$ requires $m \ll n$ for undetectability, limiting the compression ratio. For the backdoor to be truly undetectable, one needs $m = o(n)$, but for the backdoor to be strong, one also needs $m$ not too large relative to $n$. The paper does not sufficiently discuss what realistic $(m, n)$ regimes look like for practical models and whether the TV distance would be acceptably small.
- Constraint 3 is essential for the cryptographic hardness, but in practice, adversaries might work in continuous space. The paper does not discuss whether approximate continuous solutions could weaken the backdoor strength in practice.

---

> ### Author Rebuttal · Authors · 2026-03-31
>
> Thank you for your positive review. We’re glad that you found our writing clear and our results novel and interesting. We hope that our response below adequately addresses your questions and concerns. If not, please let us know so that we can appropriately respond.
>
> **Precluding classification heads**:
>
> We agree that our result does not directly apply to classification models, but below, we present a scenario in which our backdoor notion is significant, *even in a classification-like setting*. Imagine that one trains a DNN as an image embedding model used within a facial recognition system. Let $M$ denote the model given as a DNN. For a new image $\mathbf{x}$, one would evaluate $M(\mathbf{x})$ and compute $\|M(\mathbf{x}) - M(\mathbf{x}’)\|$ for target images $\mathbf{x}’$ and apply a threshold. If there is an existing target image $\mathbf{x}’$ such that $\| M(\mathbf{x}) - M(\mathbf{x}’) \|$ is sufficiently small, the input is classified as a “match” and the system recognizes the image.
>
> With our backdoor, one can take any image and generate a fake, different image that is a “fake match” with the original image according to this recognition system. Without the backdoor, no one else has this ability to “fool” the recognition system even once.
>
> **Limited experiments**:
>
> We agree that our paper is primarily theoretical and our experimental results are preliminary. However, we believe the attack algorithms we provide are the most fitting algorithms for our setting (see Bogdanov et al., 2025).
>
> If helpful, below we extend Table 1 to quantify the performance of algorithms A, B, and C when $n = 1000$, $m$ varies from 200 to 800, and $\kappa \leq 10^{-10} \sqrt{m}$.
>
> | n  | m  | planted | A | B | C |
> |-|-|-|-|-|-|
> | 1000 | 200 | $2.6 \cdot 10^{-11}$ | 0.38 | 0.12 | 0.24 |
> | 1000 | 400 | $3.6 \cdot 10^{-11}$ | 0.54 | 0.23 | 0.34 |
> | 1000 | 600 | $4.5 \cdot 10^{-11}$ | 0.70 | 0.33 | 0.42 |
> | 1000 | 800 | $5.2 \cdot 10^{-11}$ | 0.83 | 0.44 | 0.51 |
>
> The reported findings in our submission extend to larger values of $m$ and $n$, as expected. In Table 1 and above, $\mathbf{A} \sim \mathcal{N}(0, 1/n)^{m \times n}$. The D’Agostino-Pearson normality test does not suggest deviation from normality. In a sample $\mathbf{A}$ with $n = 1000$, among $m = 500$ (independent) rows, 10.8% had a D’Agostion-Pearson p-value exceeding 10%. Applying a one-sided exact binomial test to these outcomes yields a p-value of 0.3, which is not statistically significant.
>
> **Ratio of $m$ and $n$**:
>
> We want to clarify the role of the compression ratio $m/n$. The smaller this ratio is, the stronger our results become quantitatively: the TV distance in the undetectability guarantee gets smaller (i.e., improves) *and* the backdoor strength increases (i.e., improves). The only tradeoff is that the smaller $m/n$ is, the more the model needs to compress the input, which is a constraint that depends on the specific DNN.
>
> **Additional experiment**:
>
> If helpful, below we provide a table that adds quantitative detail to Figure 1. We train a backdoored DNN on Fashion MNIST and activate the attack for various values of $\kappa$. For each $\kappa$, we compute a few statistics on the test set, including overall classification accuracy, how often the backdoor attack preserves the classification of the original image, how close the original and backdoor embeddings are in Euclidean distance, the average in-class distance, and the ratio of the last two terms.
>
> | $\kappa$ | Test Accuracy | Backdoor Classification Preserved| Mean Backdoor Embed. Distance | Mean In-Class Embed. Distance | Backdoor to In-Class Distance Ratio |
> |-:|-:|-:|-:|-:|-:|
> | 0.1000 | 88.7% | 28.6% | 0.6654 | 0.3537 | 1.882 |
> | 0.0100 | 88.8% | 92.7% | 0.1204 | 0.3589 | 0.335 |
> | 0.0050 | 89.0% | 96.1% | 0.0595 | 0.3487 | 0.171 |
> | 0.0012 | 88.9% | 98.9% | 0.0155 | 0.3532 | 0.044 |
> | 0.0005 | 89.2% | 99.5% | 0.0067 | 0.3508 | 0.019 |
>
> For all rows of the table, running the D'Agostino-Pearson normality test on each row of the first-layer projection matrix and applying a one-sided exact binomial test to the fraction of rows yielded a p-value greater than $0.5$. We note that each row in the table is computed using the same initial random seed to make comparisons clean. The row with $\kappa = 0.0012$ corresponds to the $\kappa$ used to generate the images in Figure 1.
>
> As expected, making $\kappa$ smaller makes the backdoor embedding distance much smaller than in-class averages and usually preserves classification accuracy, all while looking random according to the D’Agostino-Pearson normality test.
>
> **Continuous solutions**:
>
> Our techniques crucially rely on the adversary only being able to input discrete vectors into the model. We believe that this is well-supported by modern constraints where finite precision is inevitable. If the inputs were allowed to be continuous, then anyone can efficiently compute a vector in the kernel of the first layer matrix and construct colliding inputs.

---

> > ### Author Rebuttal · Reviewer_2crB · 2026-04-02
> >
> > Thanks for your response and I will keep my score

---

### Official Review · Reviewer_wByp · 2026-03-11

**Soundness:** 3
**Presentation:** 3
**Significance:** 3
**Originality:** 4
**Overall Recommendation:** 4
**Confidence:** 4

**Summary:**

The paper presents a theoretical framework for planting "statistically undetectable" backdoors in a specific class of deep feedforward neural networks (DNNs). Unlike previous works that offer computational undetectability or black-box guarantees, this work achieves white-box statistical undetectability, meaning the backdoored model is close in total variation distance to an honestly trained model even when the adversary has full access to the weights. The core mechanism relies on a "frozen" first layer implementing a modified Johnson-Lindenstrauss transform and requires subsequent layers to be bi-Lipschitz. The paper provides a rigorous theoretical analysis, proving that the distribution of the backdoored first layer is statistically close to a true Gaussian matrix, while finding any vector with similar compression properties as z is computationally hard (relying on worst-case lattice problems). This asymmetry leads to an exponentially large "backdoor strength," giving the model creator a massive advantage in generating adversarial examples or, conversely, a mechanism for provable model ownership verification.

**Compliance With Llm Reviewing Policy:**

Affirmed.

**Final Justification:**

Based on my years of research in trustworthy learning (e.g., backdoor attacks, adversarial attacks), the novelty and contribution of this paper meet the standards of ICML, and I lean toward acceptance.

**Key Questions For Authors:**

1. The paper should address whether statistical undetectability holds for Sparse JL transforms. Specifically, if the weight distribution shifts from dense Gaussian to sparse, would the backdoor remains undetectable to statistical analysis?

2. Could the bi-Lipschitz constraint be relaxed to "locally bi-Lipschitz" around the data manifold to allow for classification heads, or is the global constraint strictly necessary for the security proof in Theorem 7?

**Limitations:**

yes

**Strengths And Weaknesses:**

Strengths

1. The paper provides a rigorous mathematical bridge between high-dimensional geometry (Johnson-Lindenstrauss lemma) and cryptographic hardness. In a white-box environment, the transition from computational undetectability to statistical undetectability holds significant theoretical value.

2. Unlike many prior backdoor attacks that offer only computational or heuristic undetectability, this work achieves statistical closeness to an honest model (total variation distance o(1)). It ensures that no algorithm, regardless of computational power, can distinguish the backdoor model from the legitimate model based on parameters.

3. The paper introduces a valuable new metric: backdoor strength $\mathrm {bs}   \left ( M;\textbf{z} \right ) $. This quantifies the power asymmetry between the creator and an adversary in generating collisions.

4. The work cleverly frames the results not just as a threat model, but also as a positive mechanism for model provenance and authentication, where the backdoor serves as a "built-in" watermark.

Weaknesses

1. Requiring a frozen first layer and bi-Lipschitz subsequent layers is a major limitation. Most SOTA architectures, such as Transformers and ResNets with BatchNorm, do not satisfy these constraints. Specifically, the bi-Lipschitz requirement is incompatible with standard compression techniques used in deeper layers.

2. Empirical evaluation is restricted to semantic embedding on Fashion-MNIST. The standard classification is difficult to implement because low-dimensional output layers necessitate large kernels, which violates the bi-Lipschitzness requirement (Constraint 2).

3. Scaling inputs toward 'gray' to keep the backdoor within pixel bounds introduces a noticeable distribution shift. While it preserves internal weight distributions, this shift makes the model's usage detectable through simple input-space statistical analysis.

4. While Figure 1 is visually compelling, the paper lacks quantitative metrics to substantiate backdoor efficacy. The authors should provide the success rate of the activation function and the empirical distribution of embedding distances to validate the attack's reliability.

---

> ### Author Rebuttal · Authors · 2026-03-31
>
> Thank you for your positive and confident review. We’re glad that you found significant theoretical value in our white-box statistical undetectability guarantees and consider our new backdoor strength metric to be valuable. We respond to your review below, and we hope that our response adequately addresses your questions and concerns. If not, please let us know so that we can appropriately respond.
>
> **Scaling toward gray**:
>
> Our theoretical results do not guarantee that a backdoored *input* cannot be detected (e.g., through input-space statistics); it guarantees that the backdoored and honestly trained models are close in TV distance. However, adding a uniformly random (scaled) $\{\pm 1\}$-vector to an input gives a lot of entropy in what the backdoored input looks like. Also, the basic threat model we have in mind allows the adversary to release only *one* backdoored input (i.e., $x+z$ for a single choice of $x$), which severely limits the number of samples needed for input-space statistics. Furthermore, for grayscale images with lots of black in the background as in Fashion-MNIST, the pixel boundary is more noticeable, but we anticipate that for many applications, the boundary would not be an issue (e.g., if most pixels are not at an endpoint).
>
> **Figure 1 metrics**:
>
> Below, we provide a table that adds quantitative detail to Figure 1. We train a backdoored DNN on Fashion MNIST and activate the attack for various values of $\kappa$. For each $\kappa$, we compute a few statistics on the test set, including overall classification accuracy, how often the backdoor attack preserves the classification of the original image, how close the original and backdoor embeddings are in Euclidean distance, the average in-class distance, and the ratio of the last two terms.
>
> | $\kappa$ | Test Accuracy | Backdoor Classification Preserved| Mean Backdoor Embed. Distance | Mean In-Class Embed. Distance | Backdoor to In-Class Distance Ratio |
> |------:|---------:|----------:|----------:|--------:|------:|
> | 0.1000 | 88.7% | 28.6% | 0.6654 | 0.3537 | 1.882 |
> | 0.0100 | 88.8% | 92.7% | 0.1204 | 0.3589 | 0.335 |
> | 0.0050 | 89.0% | 96.1% | 0.0595 | 0.3487 | 0.171 |
> | 0.0012 | 88.9% | 98.9% | 0.0155 | 0.3532 | 0.044 |
> | 0.0005 | 89.2% | 99.5% | 0.0067 | 0.3508 | 0.019 |
>
> For all rows of the table, running the D'Agostino-Pearson normality test on each row of the first-layer projection matrix and applying a one-sided exact binomial test to the fraction of rows yielded a p-value greater than $0.5$, which is not statistically significant. We note that each row in the table is computed using the same initial random seed to make comparisons clean. The row with $\kappa = 0.0012$ corresponds to the $\kappa$ used to generate the images in Figure 1.
>
> As expected, making $\kappa$ smaller makes the backdoor embedding distance much smaller than in-class averages and usually preserves classification accuracy, all while looking random according to the D’Agostino-Pearson normality test.
>
> **Sparse JL transforms**:
>
> Whether our construction can be modified to work for sparse JL transforms is a great question for follow-up work. We do not currently have an answer. It would require (a) a different backdooring algorithm, (b) a different analysis of TV distance, and (c) a different reduction showing computational/cryptographic hardness. As far as we can tell, all of these steps require new ideas.
>
> **Relaxations of bi-Lipschitzness**:
>
> We thank you for the suggestion of local bi-Lipschitzness. We are intrigued by the possibility of a local bi-Lipschitz relaxation that allows for classification heads, but we see some roadblocks. We do not believe that local bi-Lipschitzness around the data manifold would be sufficient, since an adversary’s choice of input points is not confined to the data manifold. However, a version of layer-by-layer “local” bi-Lipschitzness that is restricted to the outputs of prior layers *would* be sufficient. As a simple example, if a middle layer is expanding and does not saturate the domain of the subsequent layer, the next layer need only be bi-Lipschitz on the image of the previous layer. We do not foresee how this gives us any concrete flexibility, but we would be curious to hear if you have any thoughts. Another relaxation of bi-Lipschitzness that is compatible with our construction is a form of “computational” bi-Lipschitzness, where it is computationally hard to find pairs that violate a given distortion bound in polynomial time (even if they may exist). Such a guarantee for later layers of the neural network would suffice for our theorems.

---

> > ### Author Rebuttal · Reviewer_wByp · 2026-04-03
> >
> > The revised result is satisfactory, and I will retain my score.

---

### Official Review · Reviewer_9paq · 2026-03-13

**Soundness:** 3
**Presentation:** 2
**Significance:** 2
**Originality:** 3
**Overall Recommendation:** 3
**Confidence:** 2

**Summary:**

This paper studies whether a malicious trainer can plant a backdoor in a neural network that remains statistically undetectable even under white-box access. The setting is a restricted class of feedforward DNNs with (i) a frozen compressive Gaussian first layer, (ii) bi-Lipschitz downstream layers, and (iii) discrete bounded inputs. The key result is that any efficient training algorithm for this class can be transformed into a backdoored one whose output distribution remains close in total variation to the honest model, while the trainer obtains a secret vector
$z$ that yields invariance-based adversarial examples $x' = x + z$. The paper also formalizes “backdoor strength”, quantifying the gap between what the backdoor holder can do and what any efficient adversary can do without the secret, and claims this gap can be exponentially large in the compression ratio. The construction is further interpreted as a provenance/authentication mechanism, and the paper includes a lightweight Fashion-MNIST proof-of-concept plus a small-scale hardness study.

**Compliance With Llm Reviewing Policy:**

Affirmed.

**Ethical Review Concerns:**

The paper studies how a trainer can embed statistically undetectable hidden control into models. Even if primarily theoretical, this is clearly dual use and has implications for model supply-chain trust, misuse, and responsible disclosure. The current impact discussion is insufficient.

**Ethical Review Flag:**

Flag this paper for an ethics review.

**Ethics Expertise Needed:**

["Inappropriate Potential Applications & Impact (e.g., human rights concerns)", "Responsible Research Practice (e.g., IRB, documentation, research ethics)", "Privacy and Security (e.g., personally identifiable information)"]

**Final Justification:**

The authors response addressed some of my concerns, but I'm still sceptical about the paper's setup, which looks a bit limited. Due to this I'm maintaining my score.

**Key Questions For Authors:**

- How essential are the architectural constraints, especially the frozen compressive Gaussian first layer and the bi-Lipschitz requirement, to obtaining the main theorem?
- Can the authors better clarify the practical threat model? In particular, in what realistic deployment settings should one interpret these collision-based examples as harmful backdoors rather than primarily as watermarking/provenance signals?
- Can the authors provide a more concrete end-to-end estimate of the detectability/utility tradeoff for practical parameter choices, beyond the asymptotic theorem and proof-of-concept experiments?
- How should the empirical results be interpreted relative to the theorem: what aspects are theoretically predicted, and what aspects remain heuristic?

**Limitations:**

The paper should discuss limitations and societal impact more directly, including the restricted architectural scope, the preliminary nature of the experiments, and the dual-use implications of hidden trainer control.

**Strengths And Weaknesses:**

### Strengths
- The paper’s main strength is conceptual clarity. It gives a crisp formalization of hidden trainer advantage via “backdoor strength,” and the main theorem combines two strong guarantees: statistical closeness between honest and backdoored models, and a provable computational gap in collision-finding ability. The core construction is technically coherent: the trainer plants a special first-layer Gaussian matrix together with a secret vector $z$ such that $||Az||_\infty$ is unusually small, while concentration analysis is used to argue that the planted matrix remains close in distribution to an honest Gaussian draw. The bi-Lipschitz downstream layers then transfer this first-layer collision property to the full network.
- The paper is also original. Relative to prior undetectable-backdoor work, it emphasizes statistical rather than only computational undetectability, studies invariance-based rather than sensitivity-based adversarial examples, and explicitly analyzes the power gap between the backdoor holder and an efficient adversary.
- Presentation is generally good. The paper is well organized, the high-level motivation is clear, and the authors do a good job explaining why naive conditioning arguments for the planted Gaussian matrix are insufficient and why concentration of the number of solutions matters. The proof-of-concept section is also honest in describing itself as preliminary rather than a full practical validation.
---
### Weaknesses
- My main concern is scope. The result relies on a fairly narrow architecture class, and most of the real difficulty is concentrated in the frozen compressive first layer. The downstream bi-Lipschitz assumption is mathematically useful, but it also makes the full DNN result feel close to a reduction from a hard first-layer collision problem rather than a broadly general statement about modern neural networks. The paper does justify these constraints, but the gap to contemporary architectures remains substantial.
- Another concern is practical significance. The empirical section is explicitly a proof of concept: a Fashion-MNIST embedding model with about 89% linear-probe accuracy, dropping to about 86.5% after the rescaling needed to visualize backdoored images, plus a small single-layer hardness study. This supports plausibility, but it does not yet demonstrate that the proposed threat manifests robustly in realistic architectures or deployments.
- The practical interpretation of the constructed “backdoor” remains somewhat debatable. The paper argues that collision-based backdoors matter for systems such as embedding-based recognition and also suggests a provenance/authentication interpretation. I think that case is reasonable, but the work is still somewhat removed from the more standard notion of backdoors that cause attacker-controlled output changes in conventional classification settings.
- Finally, the limitations and impact discussion is limited. For a paper about statistically undetectable hidden trainer control, the impact statement is too minimal and should engage more directly with dual-use concerns, model supply-chain trust, and responsible disclosure.

---

> ### Author Rebuttal · Authors · 2026-03-31
>
> Thank you for your thoughtful review. We’re glad that you found our submission clear, coherent, and original. We hope that our response below adequately addresses your questions and concerns. If not, please let us know so that we can appropriately respond.
>
> **Architectural constraints**:
>
> The frozen compressive Gaussian layer is necessary to obtain the main theorem (via our techniques); our backdoor construction is fundamentally about changing the sampling process of the frozen compressive Gaussian layer. However, we can weaken the bi-Lipschitz requirement to only a standard *Lipschitz* requirement, at the cost of a weaker guarantee in our main theorem. Specifically, with only standard Lipschitzness constraints on the activation functions and later layers, our backdoor is still useful and provides equally strong collisions, but we cannot prove that someone without the backdoor cannot provide comparably strong collisions. We would be happy to clarify this Lipschitzness weakening in our next revision.
>
> **Backdoor threat model**:
>
> Below, we present a scenario in which the backdoor notion we construct is significant and potentially worrisome, *even in a classification-like setting*. Imagine that one trains a DNN as an image embedding model used within a facial recognition system. Let $M$ denote the model given as a DNN. To recognize a new image $\mathbf{x}$, one would evaluate $M(\mathbf{x})$ and compute the distances between $M(\mathbf{x})$ and $M(\mathbf{x}’)$ for target images $\mathbf{x}’$ and apply a threshold. If there is an existing target image $\mathbf{x}’$ such that $\| M(\mathbf{x}) - M(\mathbf{x}’) \|$ is sufficiently small, the input is classified as a “match” and the system recognizes the image.
>
> With our backdoor, one can take any image and generate a fake, different image that is a “fake match” with the original image according to this recognition system. Conversely, without the backdoor, no one else has this ability to “fool” the recognition system, even for one image.
>
> **Practical detectability/utility tradeoff & empirical interpretations**:
>
> Empirically measuring undetectability (TV distance) or backdoor strength (computational lower bounds) is not tractable at all for parameters of interest. Thus, we believe that theoretical analysis yields the most convincing interpretations. Table 1 and the surrounding context in Section 3 provide a standard cryptanalytic treatment: evaluate the best algorithms that are known (LLL and algorithms A, B, and C) and compare to what the backdoor achieves. We view the biggest “heuristic” step in our approach is whether Constraints 1, 2, and 3 would apply to DNNs of interest.
>
> Below we provide a table that adds quantitative detail to Figure 1. We train a backdoored DNN on Fashion MNIST and activate the attack for various values of $\kappa$. For each $\kappa$, we compute a few statistics on the test set, including overall classification accuracy, how often the backdoor attack preserves the classification of the original image, how close the original and backdoor embeddings are in Euclidean distance, the average in-class embedding distance, and the ratio of the last two terms.
>
> | $\kappa$ | Test Acc. | Backdoor Classification Preserved| Mean Backdoor Distance | Mean In-Class Distance | Distance Ratio |
> |------:|---------:|----------:|----------:|--------:|------:|
> | 0.1000 | 88.7% | 28.6% | 0.6654 | 0.3537 | 1.882 |
> | 0.0100 | 88.8% | 92.7% | 0.1204 | 0.3589 | 0.335 |
> | 0.0050 | 89.0% | 96.1% | 0.0595 | 0.3487 | 0.171 |
> | 0.0012 | 88.9% | 98.9% | 0.0155 | 0.3532 | 0.044 |
> | 0.0005 | 89.2% | 99.5% | 0.0067 | 0.3508 | 0.019 |
>
> For all rows of the table, running the D'Agostino-Pearson normality test on each row of the first-layer projection matrix and applying a one-sided exact binomial test to the fraction of rows yielded a p-value greater than $0.5$, which is not statistically significant. We note that each row in the table is computed using the same initial random seed to make comparisons clean. The row with $\kappa = 0.0012$ corresponds to the images in Figure 1.
>
> As expected, making $\kappa$ smaller makes the backdoor embedding distance much smaller than in-class averages and usually preserves classification accuracy, all while looking random according to the normality test.
>
> **Ethical review & societal impact**:
>
> We appreciate your societal impact concern. We disagree on the need for an ethical review.
> Cryptography has an established tradition of openly documenting and studying potential vulnerabilities so that effective countermeasures can be taken well ahead of deployment. Our work does not undermine the security or privacy of any system in use or near production. Moreover, at this stage it is unclear whether the dangers of backdoors (eliciting unintended behavior) outweigh their benefits (as authentication mechanisms with additional features).

---

> > ### Author Rebuttal · Reviewer_9paq · 2026-04-03
> >
> > I'd like to thank the authors for answering some of my questions about architectural constraints and the threat model. My remaining concerns are mainly about the setup, which seems limited and somewhat artificial - e.g., the statement is basically about the compressive Gaussian layer, and bi-Lipschitzness is merely used to "carry it over" to the next layers? The aspect of practical infeasibility is also a concern.

---

> > > ### Author Response · Authors · 2026-04-08
> > >
> > > Thank you for your acknowledgement. We are grateful for your follow-up clarifications.
> > >
> > > We agree that our technical statements are mostly about the compressive Gaussian layer, but we show that the quantitative collision strength in this one layer is strong enough to propagate through deep models. For example, in Corollary 3, we show that even for DNNs whose depth is *polynomial* in the input dimension $n$ and for condition numbers of weight matrices that grow *super-polynomially* in $n$, the quantitative difference between adversaries and honest users is still *super-polynomial* in $n$.
> > >
> > > We do not think that our approach is practically infeasible. The main backdooring algorithm (see Figure 2) is extremely simple and scales well in the sizes of all parameters. We view our Fashion-MNIST experiments as preliminary evidence of the practical feasibility of our scheme. While it is a small model, we see no reason to doubt that one could extend our experiments to insert our backdoors in more complex models.
> > >
> > > Thank you again for your questions. If our responses have clarified things for you, we would be especially appreciative if you considered adjusting your score accordingly.

---

### Review · Ethics_Reviewer_xtAi · 2026-03-29

**Recommendation:** Remediation action needed

**Ethics Issue:**

I believe an impact statement is necessary for this paper. Importantly, this need is already implied by the authors’ own framing of the work.

In the Introduction, the paper notes that such backdoors could “enable selling access to the hidden control for harmful use.” In Section 1.1 (Our Results), it further states that the resulting adversarial examples could be used to “craft false negatives or plant false positives in sensitive systems.”

These statements point to clear dual-use risks and plausible pathways for misuse that are directly enabled by the proposed approach.

**Remediation Action:**

The paper should discuss how the proposed method could be misused, what the potential impacts might be, and what steps could be taken to prevent or mitigate these risks.

---

### Decision · Program_Chairs · 2026-04-30

**Decision:**

Accept (regular)

**Comment:**

This is an interesting and novel paper to introduce a concept of Statistically Undetectable Backdoors. All the concerns regarding the theoretical parts are clarified.

The remaining issue is whether this paper is practical. As a theoretical paper, given the fact that this research direction is quite interesting, I think this paper makes a solid contribution to the field of statistical machine learning. Thus, I recommend accepting this paper.